# HilbertA: Hilbert-Curve-Aligned
# Sparse Attention for 2D Structured Data

**Shaoyi Zheng** [* 1] **Wenbo Lu** [* 1] **Yuxuan Xia** [1] **Shenji Wan** [1]

## Abstract

Designing sparse attention for 2D image data in diffusion and vision-language models requires reconciling spatial locality with hardware-efficient execution: handcrafted 2D sparsity patterns preserve spatial structure but often induce uncoalesced memory access, limiting practical speedups on modern GPUs. We present **HilbertA**, a 2D-aware sparse attention mechanism that reorders image tokens along a Hilbert curve, converting local spatial neighborhoods into contiguous memory segments for efficient GPU execution. To enable communication beyond local tiles, HilbertA shifts attention windows along the Hilbert-ordered sequence across layers and uses a small central shared region, preserving contiguous access while supporting cross-tile information flow. Across diffusion and vision-language models, HilbertA delivers consistent efficiency gains while maintaining competitive quality, achieving up to $4.16\times$ attention acceleration and $1.44\times$ end-to-end speedup on `Flux.1-dev`, and up to $2.30\times$ attention acceleration with $1.57\times$ faster time-to-first-token on `Qwen3-VL-8B` inference.

## 1. Introduction

Transformer-based models, widely used in modern deep-learning systems, introduce quadratic complexity in the number of tokens, leading to substantial inference latency as sequence length scales. Sparse attention mitigates this issue by restricting interactions to structured subsets of tokens and has been extensively studied in one-dimensional sequence modeling. Representative approaches such as sliding windows (Zhang et al., 2025b) and block-wise sparsity patterns (Beltagy et al., 2020) reduce computation and mem-

ory while largely preserving modeling capacity.

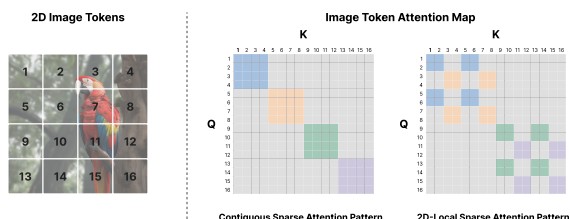

*Figure 1.* Image token layouts with their corresponding sparse attention patterns. Left: Image token layouts. Sparse patterns (Middle: Contiguous but not 2D-local; Right: 2D-local but not contiguous).

Prior work on sparse attention has largely been shaped by properties in one-dimensional sequence modeling (i.e., text), leaving two-dimensional cases such as images underexplored. In vision, an effective sparse pattern should preserve each token's two-dimensional neighborhood (a property we denote as *2D locality*) *and* align with GPU memory layouts. This creates a fundamental *dilemma* (Figure 1): memory-contiguous layouts are hardware-efficient but disrupt spatial proximity, whereas 2D-local patterns preserve spatial structure but typically induce uncoalesced memory access. For example, Swin Attention (Liu et al., 2021) and CLEAR (Liu et al., 2024a) both enforce two-dimensional locality through window-based or circular receptive fields, which are effective for preserving spatial structure. However, we argue that such 2D-local patterns typically rely on uncoalesced memory reads under standard GPU layouts, as spatially adjacent tokens are not contiguous in memory. Likewise, Sparge Attention (Zhang et al., 2025a) leverages a Hilbert curve to compress self-similar blocks, but its resulting attention patterns still induce inefficient memory layouts. In all these cases, the attention mechanisms are 2D-aware but not hardware-aligned, and low-level memory inefficiencies ultimately limit the achievable speedups in practice.

To resolve this tension between 2D locality and hardware-efficient execution, we introduce **HilbertA**, a hardware-aligned sparse attention mechanism that preserves spatial neighborhoods while minimizing uncoalesced GPU memory access. HilbertA combines three complementary components as shown in Figure 2. (i) *Reordering*: Tokens are reordered along a Hilbert curve so that spatial neighbors

---
[*]Equal contribution [1]Department of Computer Science, New York University. Correspondence to: Shenji Wan <sw5973@nyu.edu>.

*Proceedings of the 43$^{rd}$ International Conference on Machine Learning*, Seoul, South Korea. PMLR 306, 2026. Copyright 2026 by the author(s).

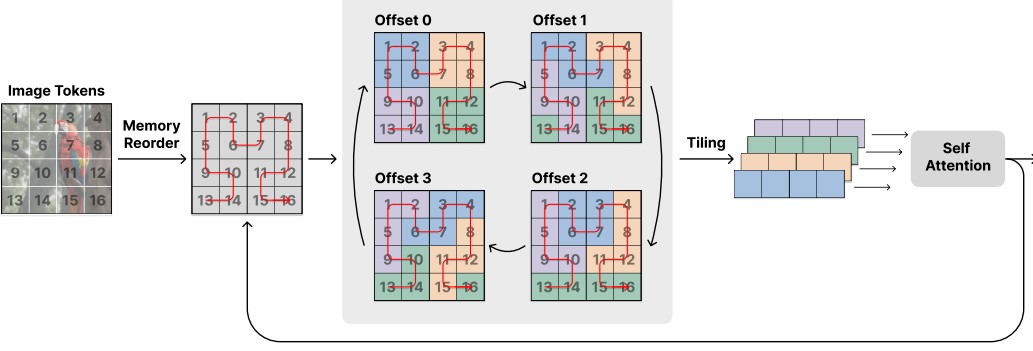

*Figure 2.* The HilbertA pipeline has three stages: (1) **Reordering** tokens along a Hilbert curve for contiguous memory layout; (2) **Tiling** tokens into local blocks for efficient intra-tile attention; and (3) **Sliding** the window by a fixed offset, enabling cross-tile interaction while preserving efficient memory access.

remain adjacent in memory. This choice is quantitatively supported by proposed metrics, showing that Hilbert curves consistently preserve spatial locality while reducing geometric distortions compared to alternative space-filling curves. (ii) *Tiling*: the reordered sequence is partitioned into tiles, within which dense attention captures fine-grained local features. This tiling accelerates attention by enforcing sparsity, while the fractal, self-similar nature of Hilbert curves ensures that tiles at different scales remain spatially coherent. (iii) *Sliding*: HilbertA employs a fixed-offset, layer-wise sliding strategy together with a static central shared region to enable structured cross-tile communication. Unlike naive pattern switching, this approach preserves the memory layout and avoids costly reallocations.

To demonstrate generality across model paradigms, we evaluate HilbertA on diffusion-based image generation and autoregressive vision-language inference:

On `Flux.1-dev` (Labs, 2024), HilbertA achieves up to $2.3\times$ attention speedup and $1.10\times$ end-to-end speedup at $1024 \times 1024$ resolution, scaling to $4.16\times$ and $1.44\times$ respectively at $2048 \times 2048$. On `FLUX.2-Klein-9B` (Labs, 2025) with 10 denoising steps, HilbertA delivers up to $1.48\times$ attention speedup and $1.07\times$ end-to-end speedup, while improving FID from 26.39 to 25.67. Notably, these gains require only lightweight LoRA fine-tuning on substantially less training data than CLEAR, yet still yield a stronger quality–efficiency trade-off. We attribute this to the locality induced by Hilbert ordering, which lowers the adaptation cost and makes HilbertA a practical sparse-attention drop-in for diffusion backbones.

On the vision-language model `Qwen3-VL-8B` (Bai et al., 2025), HilbertA achieves over $2.30\times$ attention acceleration and a $1.57\times$ improvement in time-to-first-token, while maintaining accuracy across multiple benchmarks. Across all settings, HilbertA preserves output quality—quantitatively and qualitatively—on par with strong baselines, demonstrat-

ing a favorable efficiency–fidelity trade-off.

Overall, this work makes the following contributions:

- We identify a fundamental dilemma in sparse attention for image data: preserving 2D spatial locality inherently conflicts with memory-efficient execution.

- We propose **HilbertA**, a Hilbert-curve-guided sparse attention mechanism that integrates token reordering, tiling, and layer-wise sliding to preserve 2D locality while maintaining contiguous memory access.

- We demonstrate the effectiveness and generality of HilbertA across multiple diffusion backbones and vision-language models, achieving substantial acceleration at low sparsity while maintaining comparable image quality and model performance.

## 2. Related Work

**Kernel-based and Linearized Attention.** Performer (Choromanski et al., 2020) and Linformer (Wang et al., 2020) introduce kernelization or low-rank factorization to approximate attention with linear complexity. Performer achieves linear complexity through positive orthogonal random features, while Linformer reduces computation by projecting sequences into lower-dimensional spaces. These approaches are complementary to ours.

**Sparse Attention Patterns.** Sparse attention mechanisms reduce computational complexity by limiting token-to-token interactions. Early methods such as Sparse Transformer (Child et al., 2019) and Longformer (Beltagy et al., 2020) adopt fixed windowed or local+global attention patterns, while BigBird (Zaheer et al., 2020) extends these with block-sparse, random, and global connections to enhance model capacity. MInference (Jiang et al., 2024)

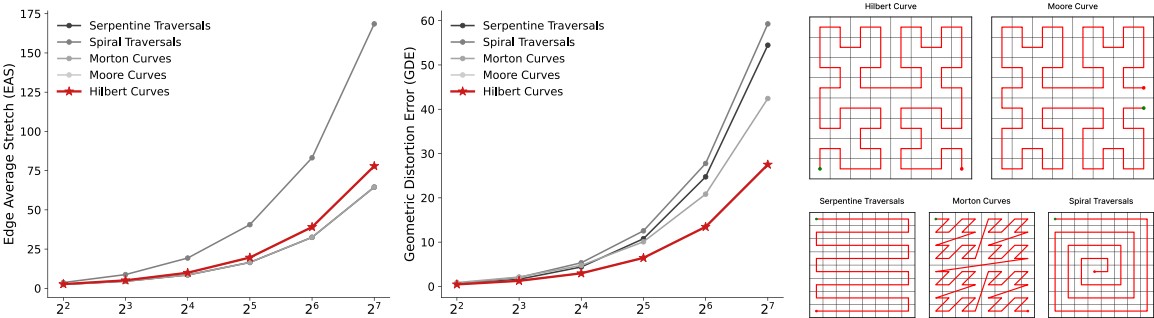

*Figure 3.* Comparison of space-filling curves. Left and middle: Edge Average Stretch (EAS) and Geometric Distortion Error (GDE) across grid sizes. Hilbert curves achieve the lowest GDE and competitive EAS compared with serpentine, Morton, Moore, and spiral orderings. Right: example orderings on a $2^3$ grid.

proposes prefill sparsity by reducing attention computation to only critical regions. NSA (Yuan et al., 2025) from DeepSeek introduces a trainable sparse mask tailored for prefill acceleration. XAttention (Xu et al., 2025) adaptively selects blocks based on anti-diagonal saliency to balance compute and performance. MoBA (Lu et al., 2025) dynamically merges attention blocks guided by similarity-based objectives, improving routing efficiency across token clusters. However, despite growing interest in sparse attention for language models, few methods have explicitly explored 2D-aware sparse patterns tailored for image generation tasks, where spatial structure is critical.

**I/O-Aware Tiled Attention.** FlashAttention (Dao, 2023) accelerates exact attention by computing in small tiles that fit into fast memory, thereby minimizing memory I/O without sacrificing accuracy. This approach demonstrates how careful memory management can substantially reduce computational overhead. Our method is related in spirit but differs in focus: rather than tiling the attention computation itself, Hilbert Attention enhances efficiency through memory layout optimized by Hilbert-curve token reordering, which preserves spatial locality while ensuring coalesced access. In this sense, our method complements FlashAttention by addressing efficiency from another angle.

**Hilbert Curve.** Hilbert curves are widely used to preserve locality when mapping 2D data to 1D, and have shown benefits for neighborhood structure and coherence in classification and spatial modeling tasks (Erkan & Aksoy, 2023; Tang et al., 2023). SpargeAttention (Zhang et al., 2025a) extends Hilbert curves into sparse attention but only as a pre-processing step, with sparsity ultimately dictated by dynamic similarity thresholds; the memory access remains uncoalesced. In contrast, our method directly designs the sparse pattern and propagates information following Hilbert curves. This design preserves contiguous memory access, translating naturally into hardware efficiency.

## 3. Method

We propose a lightweight mechanism, *Hilbert Attention(HilbertA)*, that simultaneously ensures spatial locality, efficient information propagation, and reduced memory cost. The design integrates three components: Hilbert-curve reordering for contiguous token access, two-dimensional tiling for local attention, and cross-layer sliding to enable structured communication across tiles.

### 3.1. Space-Filling Curves for Token Ordering

Transformers operate on one-dimensional sequences, whereas images are naturally arranged on two-dimensional grids. This mismatch requires us to "reorder" image tokens into a sequence. Under full attention, the ordering is immaterial because diffusion Transformers are permutation equivariant. But order becomes critical with sparse attention, where each token interacts only within a restricted neighborhood of tokens. In this case, the choice of linearization determines which neighbors fall into the same attention block. Poor ordering can scatter nearby patches across blocks and weaken the model's ability to capture local structure. In contrast, a good ordering preserves the spatial locality of images, making sparse attention more effective.

To make these considerations precise, we model the image grid as a graph $G = (V, E)$, where $V = \{(i, j) \mid 0 \le i < m, \ 0 \le j < n\}$ are the $m \times n$ cells and $E \subseteq V \times V$ connects 4-neighbor adjacencies. Let $N = mn$. A candidate linearization is a bijection $\pi : \{0, \ldots, N-1\} \to V$ that visits each cell exactly once, with inverse $\pi^{-1}(v)$ giving the position of $v$ in the sequence. The first metric we consider is the *Edge Average Stretch (EAS)*, which measures how far immediate grid neighbors are separated after linearization. Lower EAS indicates better preservation of local adjacency. Formally,

$$\text{EAS}(\pi) = \frac{1}{|E|} \sum_{(u,v) \in E} d_{1\text{D}}(u, v),$$

where $E$ denotes the set of four-neighbor edges and $\pi^{-1}(u)$ is the position of cell $u$ in the one-dimensional sequence.

$$d_{1\text{D}}(u,v) = \left| \pi^{-1}(u) - \pi^{-1}(v) \right|,$$

Up to the normalization factor $|E|$, this is exactly the total edge length objective studied in the Minimum Linear Arrangement problem (Mitchison & Durbin, 1986).

However, EAS only measures local adjacency preservation. A layout can minimize total edge length while still producing a globally unbalanced or anisotropic ordering. For example, the optimal grid arrangement of Mitchison & Durbin (1986) contains large row-wise and column-wise regions and is highly asymmetric; although optimal under the MinLA objective, such a layout is not ideal for sparse attention, where contiguous windows should correspond to compact and evenly distributed two-dimensional tiles.

This motivates a second metric that evaluates whether the full one-dimensional geometry remains consistent with the original two-dimensional geometry across scales.

$$GDE(\pi) \;=\; \frac{1}{M} \sum_{(u,v)\in S} \Big( \alpha(\pi)\, d_{1\text{D}}(u,v) - d_{2\text{D}}(u,v) \Big)^2,$$

where $M = |S|$, denoting the summation over a chosen set of pairs $S \subseteq V \times V$, and with $u = (i_u, j_u)$ and $v = (i_v, j_v)$ indicating their grid coordinates:

$$\alpha(\pi) = \frac{\sum_{(u,v)\in S} d_{1\text{D}}(u,v)\, d_{2\text{D}}(u,v)}{\sum_{(u,v)\in S} d_{1\text{D}}(u,v)^2},$$

$$d_{2\text{D}}(u,v) = \big\| (i_u, j_u) - (i_v, j_v) \big\|_2.$$

A low GDE indicates that the ordering provides a consistent, globally scaled embedding of the 2D geometry. Low distortion is crucial in diffusion transformers, as maintaining proportional distances allows the attention to capture both fine-grained textures and broader structural patterns.

As shown in Figure 3, we evaluate several candidate orderings using the proposed metrics. Serpentine and spiral traversals are simple to construct but introduce long jumps that inflate both edge average stretch and geometric distortion error. Morton (Z-order) curves improve hierarchical coherence, yet tend to fragment local neighborhoods. By contrast, Hilbert curves consistently achieve lower stretch and minimal distortion across grid sizes, owing to their fractal structure. We further provide a detailed empirical and qualitative comparison of different scanning curves, including serpentine and Morton orderings, in Appendix H.

## 3.2. Hilbert Curve for General Shape

While the classical Hilbert curve is typically defined over square grids with even (often power-of-two) dimensions, HilbertA adopts a generalized Hilbert curve construction that naturally supports arbitrary shapes, including non-square and odd-sized spatial dimensions, as shown in Figure 4.

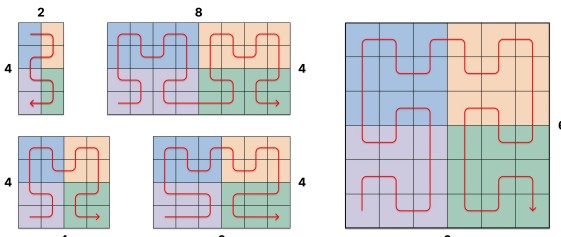

*Figure 4.* Generalized Hilbert curves on non-power-of-two grids. We show five examples: 2×4, 8×4, 4×4, 6×4, and 6×6. In all cases, the curve visits every cell exactly once with orthogonal steps, and spatial neighbors remain close in the 1D sequence order.

This generalized formulation allows token reordering to be defined directly on the original feature map without requiring padding to power-of-two resolutions. By operating on a generalized Hilbert ordering, HilbertA preserves spatial locality across arbitrary shapes while maintaining a regular and contiguous memory layout. This design remains GPU-friendly, as it avoids irregular indexing, reshaping, or copying, and enables coalesced memory access during attention computation across all supported spatial configurations.

These results provide a principled justification for employing Hilbert curves to reorder image tokens in DiTs. Their ability to preserve spatial locality while reducing geometric distortion makes them especially well-suited to sparse attention, resulting in more accurate and coherent generations.

## 3.3. Tile and Slide

For attention mechanisms over two-dimensional visual tokens, sparse patterns must jointly satisfy three objectives: (1) preserving fine-grained local structure, (2) enabling global information propagation, and (3) maintaining computational and memory efficiency. To address these requirements, we introduce a *tile-and-slide* mechanism:

**Local tiling for sparse attention.** Given a Hilbert-reordered sequence, we partition it into non-overlapping tiles of $N_T$ tokens and restrict attention within each tile. This strategy preserves the 2D spatial locality of images while bounding computations and memory reads to the tile area rather than the full image, improving efficiency without sacrificing local fidelity. The tile size can be flexibly chosen: smaller tiles enforce stronger locality priors and increase sparsity for higher efficiency, whereas larger tiles

provide richer context and better capture longer-range spatial structure. Crucially, because the Hilbert Curve is fractal and self-similar, tiles of any size remain spatially coherent, ensuring they form well-structured neighborhoods in 2D.

**Sliding for information propagation.** Local partitioning alone isolates tokens within tiles, blocking information propagation, and straightforward fixes such as overlapping windows (e.g., CLEAR) or cross-tile attention (e.g., SpargeAttention) typically lead to uncoalesced memory access and reduced efficiency. To address this, we adopt a sliding schedule along the Hilbert-ordered sequence, which preserves contiguous reads while enabling structured cross-tile information exchange.

Formally, let $N$ denote the sequence length after Hilbert reordering, $N_T$ the tile size in tokens, and $T = N/N_T$ the number of tiles. We define tiles as

$$\text{tile}[q] = [\, qN_T, \, (q+1)N_T \,), \quad q \in \{0, \ldots, T-1\}.$$

Choose a sliding cycle length $L \in \mathbb{N}$ and set the per-layer advance $\Delta = N_T/L$. Each layer advances the attention window by $\Delta$ tokens, so after $L$ layers the window has shifted by one tile ($N_T$ tokens), completing a cycle. An example of such a cycle can be found in Figure 2. At layer $\ell \in \{0, 1, 2, \ldots\}$, the attention window for a query token at index $i \in \{0, \ldots, N-1\}$ is

$$\mathcal{A}_i^{(\ell)} = \text{tile}\big[q_i(\ell)\big], \quad \text{where} \quad q_i(\ell) = \left\lfloor \frac{i + \ell\Delta}{N_T} \right\rfloor \, (\text{mod } T).$$

As soon as a token from a new tile enters the window, it serves as a messenger: having aggregated the information of its original tile in the previous layer, it carries that information into the new tile. Thus, the effective receptive field (ERF)—the set of tokens that influence token $i$ through a chain of attentions—grows by an entire tile per layer rather than by $\Delta$ tokens. If $q_0 = \lfloor i/N_T \rfloor$ is the initial tile of token $i$, then after $t$ layers,

$$\text{ERF}_i(t) = \bigcup_{k=0}^{\min\{T, t\}-1} \text{tile}\big[(q_0 + k) \, (\text{mod } T)\big],$$

with size and coverage ratio

$$|\text{ERF}_i(t)| = N_T \cdot \min\{T, t\},$$

$$C_i(t) = \frac{|\text{ERF}_i(t)|}{N} = \min\Big\{ 1, \frac{t}{T} \Big\}.$$

In the example of Figure 5, we set $L = 4$ and $N_T = 4$, so $\Delta = 1$: the window visits four adjacent, two-dimensional neighbor tiles per cycle while maintaining coalesced, contiguous memory access along the Hilbert order. This schedule guarantees that, after $T$ layers, the ERF spans the entire

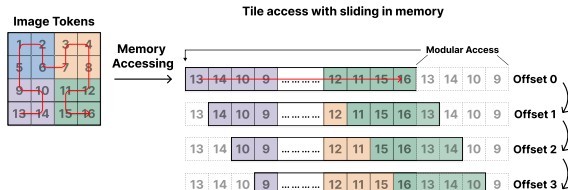

*Figure 5.* Illustration of sliding tile access along the Hilbert-ordered sequence. Left: image tokens are reordered by the Hilbert curve and partitioned into tiles. Right: modular indexing enables contiguous memory access as the attention window advances, ensuring efficient cross-tile communication without uncoalesced reads.

sequence and every token has an indirect path to all other tokens, even with local attention alone.

For maximum efficiency, we use modular indexing: when a read reaches the end of the sequence, it wraps to the head, preserving coalesced access without extra copies. This design reveals an inherent efficiency–fidelity trade-off. The tile size $N_T$ mediates locality versus global context: smaller $N_T$ improves efficiency by limiting computation to finer regions, while a larger $N_T$ enhances global consistency by capturing broader spatial interactions. By tuning $N_T$, one can balance locality, coherence, and efficiency within a fixed memory budget.

### 3.4. Shared Region for RoPE

To further enhance global coherence, we introduce a fixed shared region at the image center that every tile attends to. This region serves two complementary roles. First, it facilitates long-range communication by acting as a global relay, enabling information exchange across tiles. Second, it provides a positional anchor under Rotary Positional Embeddings (RoPE) (Su et al., 2024). While RoPE encodes relative positions within each tile, it does not convey a tile's location in the full image; a shared central region offers a reference point, improving global structural consistency.

We adopt a static central region to preserve contiguous memory access and avoid pipeline complexity. Dynamic or learned alternatives typically require per-layer reindexing, additional memory movement, or extra parameters, which undermine hardware efficiency and robustness. By contrast, a fixed shared region is parameter-free, hardware-friendly, and effective across resolutions.

### 3.5. Triton Kernel Design

With a fixed Hilbert ordering, token reordering incurs a constant overhead of two gather operations per inference and does not scale with network depth. This enables attention computation to operate directly on a contiguous memory layout throughout the network. Within this layout, computation is primarily confined to tile-local regions, with limited inter-

action to a small set of shared tokens (e.g., prompts or shared regions). Moreover, information propagation across tiles is enabled through the sliding mechanism implemented as index shifting inside the kernel rather than explicit memory movement. By advancing attention windows while keeping the underlying memory layout unchanged, this design preserves contiguous access and avoids costly data copies.

# 4. Experiments

## 4.1. Experiment Setup

**Setup.** We evaluate HilbertA on both diffusion models and vision-language models. For image generation, we evaluate HilbertA on two diffusion backbones: `Flux.1-dev` and `FLUX.2-Klein-9B`. For Flux.1, we generate images at resolutions of 1024×1024 and 2048×2048 with 35 denoising steps. For FLUX.2, we evaluate 1024×1024 image generation with 10 denoising steps. Image quality is assessed using standard metrics including LPIPS (Zhang et al., 2018), CLIP-I (Radford et al., 2021), and FID (Heusel et al., 2017). Unless otherwise specified, CLIP-I and LPIPS are computed against the corresponding dense-model outputs, and FID is computed against COCO Test-17k (Lin et al., 2014) real images.

For vision-language models, we evaluate HilbertA on `Qwen3-VL-8B-Instruct` and report performance on a diverse set of standard benchmarks. These include InfoVQA (Mathew et al., 2022), SEED-Bench (Li et al., 2023), MMBench (Liu et al., 2024b), OCRBench (Liu et al., 2024c), and MMStar (Chen et al., 2024). These benchmarks jointly assess visual understanding, multimodal reasoning, and OCR-centric tasks that rely on fine-grained spatial layouts, making them well-suited for evaluating the effectiveness of 2D-aware sparse attention. Each benchmark follows the official evaluation protocols.

Inference efficiency is measured by the median wall-clock latency over multiple runs on a single NVIDIA A100 GPU.

**Training.** In our experiments, we empirically find that HilbertA can be directly applied to Qwen3-VL without additional training, while diffusion models benefit from lightweight fine-tuning to better preserve image quality.

Accordingly, for diffusion models, we adapt HilbertA using LoRA fine-tuning rather than full-model training or large-scale distillation. For both Flux.1 and FLUX.2, we fine-tune LoRA modules with rank $r = 4$ using a curated subset of 10k images from the CLEAR setting, paired with high-quality textual annotations. Unless otherwise specified, we use 16 Hilbert tiles and a sliding cycle length of 4. The Flux.1 LoRA fine-tuning is conducted in two stages: 9 epochs at $1024 \times 1024$ resolution followed by 3 epochs at $2048 \times 2048$ resolution, totaling 80 GPU-hours on 4

NVIDIA H100 GPUs.[1] For FLUX.2, we construct a 10k-image fine-tuning set using the same prompts as the CLEAR dataset and generate the corresponding images with the dense model. We then fine-tune HilbertA on this generated dataset for 4 epochs at $1024 \times 1024$ resolution, costing a total of 12 GPU-hours on 2 NVIDIA A100 GPUs.

To further assess whether HilbertA introduces any training instability, we additionally conduct a **train-from-scratch** experiment on LightningDiT-B/1 (VAVAE-f16d32) (Yao et al., 2025). The model trains stably and converges comparably to the baseline, achieving competitive image quality. Detailed training dynamics are provided in Appendix D.

**Baseline.** As few existing methods are explicitly designed for hardware-efficient sparse attention in image generation, we compare HilbertA with two representative diffusion baselines: CLEAR and Sparge Attention. CLEAR enforces local attention through fixed-radius circular neighborhoods with additional global aggregation, while Sparge Attention predicts block-level sparsity by compressing self-similar blocks. Although both methods incorporate two-dimensional locality, their attention patterns induce uncoalesced memory access, making them suitable for comparison.

For VLM, we compare HilbertA against the Swin Transformer, a canonical baseline for window-based sparse attention in multimodal architectures. Both Swin and HilbertA aim to preserve two-dimensional spatial locality under sparse attention and rely on pattern switching across layers to enable information propagation, but Swin does not maintain contiguous memory access during window switching and incurs additional system overhead. Since Swin attention is not supported in diffusion model architectures and therefore is not included in our main diffusion experiments, for completeness, we implement a minimal prototype to compare Swin-style attention with HilbertA in diffusion models, and report both quantitative results and a detailed analysis of their methodological differences in Appendix E.

## 4.2. Ablation Studies

We further ablate two key components of HilbertA: the sliding cycle and the shared region. For the sliding cycle, we compare cycle lengths of 2 and 4 under the same resolutions and tile configurations. As shown in Table 2, changing the sliding cycle has only a minor effect on image quality, indicating that HilbertA is relatively insensitive to the exact cycle length once cross-tile communication is enabled. For the shared region, we observe a clearer trend: adding a central shared region consistently improves both LPIPS and CLIP-I, with $16 \times 16$ shared tokens at $1024 \times 1024$ and

---

[1]Dataset available at: https://huggingface.co/datasets/jackyhate/text-to-image-2M/resolve/main/data_1024_10K/data_000000.tar.

| Resolution | Configurations | | Sparsity | Image Quality metrics | | | GFLOPs | Latency | |
| | Method | Specs | | FID↓ | LPIPS↓ | CLIP-I↑ | | Attention(ms)↓ | End-to-End(s)↓ |
|---|---|---|---|---|---|---|---|---|---|
| 1024×1024 | Flux.1-dev | - | 0% | 30.6 | 0.0 | 100.0 | 261 | 1.42(1.00×) | 13.85(1.00×) |
| | Sparge Attention | - | 17% | 28.7 | 0.515 | 91.3 | 216 | 2.94(0.48×) | 14.32(0.97×) |
| | CLEAR | r= 8 | 95% | 33.0 | 0.500 | 90.0 | 64 | 0.92(1.54×) | 12.85(1.08×) |
| | | r=16 | 80% | 32.4 | 0.470 | 91.4 | 81 | 1.20(1.18×) | 13.31(1.04×) |
| | | r=32 | 22% | 32.2 | 0.429 | 92.7 | 154 | 1.71(0.83×) | 14.20(0.98×) |
| | HilbertA | 4 tiles | 75% | 33.5 | 0.520 | 90.5 | 88 | 0.78(1.82×) | 12.83(1.08×) |
| | | 16 tiles | 94% | 31.3 | 0.563 | 87.6 | 49 | 0.62(2.29×) | 12.57(1.10×) |
| 2048×2048 | Flux.1-dev | - | 0% | 32.6 | 0.0 | 100.0 | 3,299 | 18.14(1.00×) | 65.28(1.00×) |
| | Sparge Attention | - | 14% | 28.2 | 0.383 | 93.0 | 2,830 | 21.44(0.85×) | 78.64(0.83×) |
| | CLEAR | r= 8 | 99% | 43.2 | 0.616 | 80.5 | 246 | 4.84(3.75×) | 47.81(1.37×) |
| | | r=16 | 95% | 39.3 | 0.593 | 83.4 | 353 | 7.59(2.39×) | 52.31(1.25×) |
| | | r=32 | 80% | 36.9 | 0.591 | 83.4 | 724 | 12.55(1.45×) | 60.65(1.08×) |
| | HilbertA | 4 tiles | 75% | 38.4 | 0.515 | 84.1 | 1,115 | 7.37(2.46×) | 50.19(1.30×) |
| | | 16 tiles | 94% | 47.5 | 0.571 | 78.2 | 496 | 4.36(4.16×) | 45.19(1.44×) |
| 1024×1024 | FLUX.2-Klein-9B | - | 0% | 26.39 | 0.0 | 100.0 | 348 | 1.72(1.00×) | 4.46(1.00×) |
| | Sparge Attention | - | 71% | 28.03 | 0.645 | 82.7 | 101 | 1.39(1.24×) | 4.35(1.03×) |
| | CLEAR | r= 8 | 99% | 31.59 | 0.615 | 84.5 | 82 | 1.19(1.45×) | 4.28(1.04×) |
| | | r=16 | 95% | 29.77 | 0.647 | 83.2 | 118 | 1.54(1.12×) | 4.40(1.01×) |
| | HilbertA | 4 tiles | 75% | 26.26 | 0.516 | 90.3 | 117 | 1.20(1.43×) | 4.32(1.03×) |
| | | 16 tiles | 94% | 25.67 | 0.600 | 85.1 | 65 | 1.16(1.48×) | 4.17(1.07×) |

*Table 1.* Comparison of image quality and efficiency across methods and their specifications. We report sparsity, FID, LPIPS, CLIP-I, GFLOPs, attention latency, and denoising time. Numbers in parentheses denote speedups over the corresponding dense baseline within each resolution/backbone setting. Within each setting, the best and 2nd best values for each metric, excluding the dense baseline, are highlighted, respectively. (↑: higher better, ↓: lower better).

$32 \times 32$ shared tokens at $2048 \times 2048$ achieving the best results. These results support the role of sliding for cross-tile information exchange and the shared region for improving global coherence and positional anchoring. Additional qualitative comparisons are provided in Appendix C.

| Component | Setting | LPIPS↓ | CLIP-I↑ |
|---|---|---|---|
| Sliding Cycle | 2 | 0.521 | 0.904 |
| | 4 | **0.520** | **0.905** |
| Shared Region | $0 \times 0$ | 0.592 | 0.871 |
| | $8 \times 8$ | 0.545 | 0.894 |
| | $16 \times 16$ | **0.516** | **0.901** |

*Table 2.* Sliding enables cross-tile communication with similar quality across tested cycle lengths, while a slightly larger yet lightweight shared region improves LPIPS and CLIP-I.

### 4.3. Efficiency and Quality Analysis

**Efficiency Analysis** We evaluate the efficiency of HilbertA against representative sparse-attention baselines, reporting both kernel execution time and end-to-end latency or time-to-first-token (TTFT).

For diffusion models, HilbertA is tested with 4 and 16 tiles and fixed shared-region sizes (256 at $1024 \times 1024$, 1024 at $2048 \times 2048$) against CLEAR and Sparge Attention. Sparge Attention relies on heterogeneous layer-wise hyperparameters and requires nontrivial tuning; we report its best-performing configuration after 11.5 GPU-hours of search.

For completeness, we additionally report a higher-sparsity variant of Sparge Attention in Appendix E. As illustrated in Figure 6, HilbertA achieves the largest efficiency gains across resolutions. At $1024 \times 1024$, HilbertA reaches up to $2.30\times$ speedup in attention latency and $1.10\times$ in end-to-end runtime, outperforming both CLEAR and Sparge Attention. At $2048 \times 2048$, the advantage further widens to $4.16\times$ and $1.44\times$, respectively. In contrast, CLEAR yields smaller gains, while Sparge Attention is frequently slower than dense attention in our setting.

For FLUX.2-Klein-9B, we further evaluate whether HilbertA transfers to a different diffusion backbone. The same illustration shows that HilbertA again achieves the strongest overall efficiency among the compared methods. With 4 tiles, HilbertA reduces attention latency from 1.72 ms to 1.20 ms and end-to-end generation time from 4.46 s to 4.32 s. With 16 tiles, the advantage further increases, reaching the lowest attention latency of 1.16 ms and the fastest end-to-end time of 4.17 s. In comparison, Sparge Attention achieves 1.39 ms attention latency and 4.35 s end-to-end time, while CLEAR-r8 obtains a similar kernel latency but a slower end-to-end time. CLEAR-r16 is slower in both attention latency and end-to-end generation. These results indicate that the efficiency gains of HilbertA are not specific to Flux.1, but generalize to FLUX.2, where the contiguous layout successfully translates sparse computation into practical wall-clock acceleration.

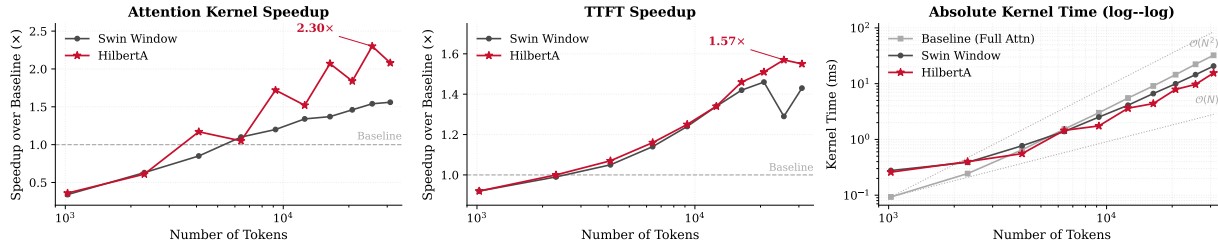

*Figure 6.* Efficiency illustration of HilbertA across diffusion and vision-language settings. By aligning sparse attention with Hilbert-ordered contiguous memory access, HilbertA converts structured sparsity into practical kernel and end-to-end speedups.

For the VLM setting, Figure 6 illustrates the results on `Qwen3-VL-8B`, where HilbertA uses 4 tiles and no shared region and is compared against the Swin transformer under a matched sparsity of 75%. HilbertA consistently improves both kernel time and TTFT over Swin across sequence lengths. Moreover, the gap becomes increasingly pronounced as the number of tokens grows: while Swin and HilbertA are similar at short contexts, HilbertA exhibits substantially better scaling at long contexts, delivering over $2.30\times$ attention acceleration and $1.57\times$ TTFT improvement at the longest sequences we test.

These results indicate that HilbertA's hardware-aligned design leads to real end-to-end gains, with the efficiency becoming more pronounced in the long-context regime.

**Overhead Analysis** HilbertA is also lightweight. Operations like tile-based sliding are naive memory index shifting and incur minimal cost. The only overhead is the token reordering. To quantify the reordering impact, Table 3 reports the latency of applying and recovering the Hilbert order, as well as its relative overhead compared to attention time. We observe that this cost remains consistently low: it constitutes only 1.9%–7.2% of total full attention time. Importantly, this reordering is applied only twice—once at the input and once at the output, and *independent* of the number of transformer layers or generation steps. In diffusion-based generation, where the model performs dozens of denoising steps (e.g., 28 or more), this one-time cost is amortized across the entire process. In summary, Hilbert-based token reordering introduces a small, bounded overhead that does not scale with model depth or generation length.

| Resolution | Tokens | Permutation Time | | Relative Overhead | | |
| --- | --- | --- | --- | --- | --- | --- |
| | | Reorder (ms) | Recover (ms) | Full Attn (%) | Tile4 (%) | Tile16 (%) |
| $1024^2$ | 4096 | 59.221 | 104.118 | 7.20 | 13.08 | 16.56 |
| $2048^2$ | 16,384 | 264.332 | 290.280 | 1.92 | 4.71 | 7.97 |

*Table 3.* Reordering and recovery overhead introduced by HilbertA, reported in absolute time and as a percentage of full attention and two sparse-attention configurations.

**Source of Acceleration** We define sparsity as the fraction of skipped entries in the attention matrix, $\text{sparsity} = 1 - \text{nnz}_{\text{pat}}(QK^\top)/N^2$, where $\text{nnz}_{\text{pat}}(\cdot)$ denotes the number of computed entries under a sparse pattern. Despite operating at lower sparsity than competing methods—which would typically be expected to yield less acceleration—and performing more FLOPs in some settings, HilbertA consistently achieves larger speedups. This observation indicates that runtime efficiency is governed not only by sparsity but also by memory access behavior. By enforcing a contiguous, hardware-aligned memory layout through Hilbert-based ordering, HilbertA enables sparse computation to translate directly into wall-clock acceleration. In contrast, methods such as CLEAR, Swin, and Sparge Attention disrupt memory contiguity through overlapping windows, pattern switching, or runtime regrouping, preventing sparsity from fully converting into end-to-end speedups.

Overall, this analysis demonstrates that effective sparse attention must be hardware-aware: reducing FLOPs alone is insufficient unless accompanied by coalesced memory access, a property that HilbertA explicitly enforces.

**Quality Analysis.** For diffusion models, HilbertA maintains a favorable quality–efficiency trade-off on both `Flux.1-dev` and `FLUX.2-Klein-9B`. On Flux.1, it matches representative sparse-attention baselines in image quality while delivering the strongest efficiency gains: at $1024 \times 1024$, the 16-tile setting reaches a competitive FID of 31.3, and at $2048 \times 2048$, the 4-tile setting preserves LPIPS and CLIP-I comparable to CLEAR with substantially lower latency. On FLUX.2, HilbertA further improves FID from 26.39 to 26.26 with 4 tiles and 25.67 with 16 tiles, while maintaining higher CLIP-I than Sparge Attention and CLEAR. Overall, smaller-tile settings preserve fidelity, whereas higher-sparsity settings provide larger speedups with moderate quality trade-offs.

Figure 7 shows qualitative comparisons at $1024^2$ and $2048^2$ across CLEAR ($r=8$), HilbertA (4 tiles, sliding cycle $L=4$), SpargeAttention, and the original Flux.1 and FLUX.2 models. HilbertA produces visually faithful samples comparable to the dense baseline while preserving its efficiency gains.

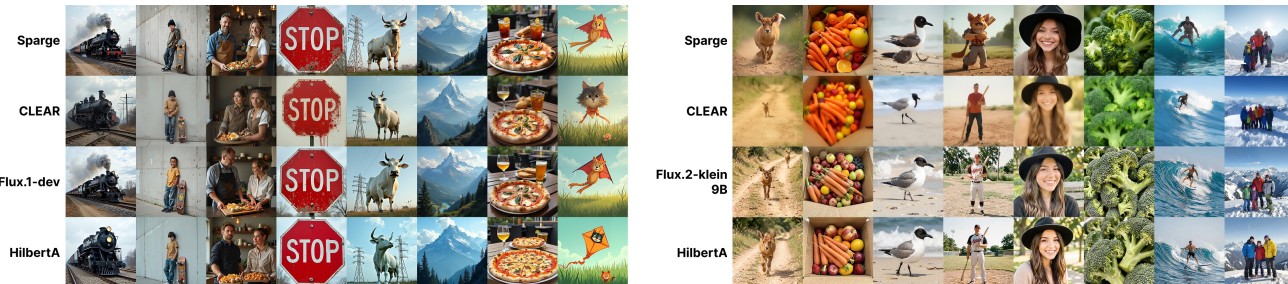

*Figure 7.* Qualitative comparison at $1024 \times 1024$ resolution on two diffusion backbones: `Flux.1-dev` on the left and `FLUX.2-Klein-9B` on the right. For each backbone, we compare the dense baseline, Sparge, CLEAR, and HilbertA. HilbertA preserves comparable visual quality across both backbones. Additional qualitative results are provided in Appendix A.

For **vision-language models**, we also observe a performance–efficiency trade-off under HilbertA. As shown in Table 4, HilbertA achieves 81.52 on InfoVQA, 0.753 on SEEDBench, 0.705 on MMBench, 686 on OCRBench, and 0.446 on MMStar, outperforming Swin attention under the same sparsity budget, while maintaining dense-level performance. Notably, this improvement is observed across all benchmarks without any single task exhibiting a disproportionate drop. We attribute this advantage to the fact that, at matched sparsity, HilbertA preserves cross-token interactions more consistently than Swin attention, whose window partitioning and pattern switching can interrupt information flow across spatially distant regions. This property is particularly beneficial for benchmarks that depend on fine-grained spatial layouts and long-range reasoning. Thus, HilbertA avoids task-specific failure modes and yields more uniform performance across diverse benchmarks.

| Method | InfoVQA | SEEDBench | MMBench | OCRBench | MMStar |
|---|---|---|---|---|---|
| Baseline | 83.09 | 0.769 | 0.737 | 912 | 0.485 |
| Swin | 79.27 | 0.737 | 0.645 | 606 | 0.404 |
| HilbertA | **81.52** | **0.753** | **0.705** | **686** | **0.446** |

*Table 4.* Performance comparison on vision-language benchmarks. Compared to Swin, HilbertA consistently achieves higher scores across tasks, while remaining competitive with the dense baseline.

**Training Budget Analysis** HilbertA also differs from prior sparse-attention adaptation methods in its training cost. We adapt HilbertA with lightweight LoRA fine-tuning, updating only about 1% of the model parameters using 10k images for four epochs. By comparison, CLEAR uses a substantially larger distillation setup, updating roughly 70% of the parameters with 750k images. Thus, CLEAR uses an approximately $18\times$ larger data budget than HilbertA. Despite this larger adaptation budget, CLEAR underperforms HilbertA on FLUX.2 across the evaluated quality–efficiency metrics. This indicates that the Hilbert-ordered locality prior provides an effective inductive bias for 2D sparse attention, allowing the model to obtain favorable efficiency–quality trade-offs without extensive retraining. We provide a detailed training-budget comparison in Appendix J.

**Summary.** Across both diffusion models and vision-language models, HilbertA consistently operates on a favorable quality–efficiency frontier, matching the quality of strong baselines while achieving substantial acceleration. Our results show that reduced sparsity or FLOPs alone do not guarantee practical speedups. By enforcing hardware-aligned sparsity, HilbertA reliably translates sparse computation into real efficiency, a property that becomes increasingly important in high-resolution and long-context regimes.

## 5. Discussion

### 5.1. Extension to Video Diffusion

HilbertA can be naturally extended to video diffusion by adopting a three-dimensional Hilbert curve that traverses the spatiotemporal volume. This preserves spatial locality within frames and temporal continuity across frames, with tiles becoming spatiotemporal blocks and sliding attention extending over both space and time. In this setting, the shared region should also be generalized to three dimensions under 3D RoPE, providing consistent spatial and temporal anchors. We leave a full exploration of this extension to future work.

## 6. Conclusion

We present Hilbert Attention, a sparse attention framework that demonstrates how 2D locality and hardware-aligned memory access can be jointly leveraged to achieve practical acceleration in vision models. By reordering tokens along a Hilbert curve, HilbertA enables sparse computation to translate into acceleration while maintaining competitive performance. Across both diffusion models and VLMs, our results highlight that sparsity alone is insufficient; efficient attention must be aligned with GPU memory access patterns to deliver speedup. By lowering the computational cost of large-scale vision and vision-language inference, HilbertA improves the practicality of deploying models in long-context and high-resolution settings.

## Impact Statement

This paper studies hardware-efficient sparse attention mechanisms for accelerating high-resolution and long-context generative models. By improving inference efficiency and reducing computational cost, our work may help lower the barrier to deploying diffusion and vision-language models in resource-constrained environments, enabling broader access to advanced generative technologies in domains such as scientific imaging, healthcare, and education.

At the same time, increased efficiency can also make large-scale generative models more accessible for malicious or irresponsible use, including the generation of misleading or harmful content. These risks are not unique to our method, but are inherent to generative models more broadly. Addressing such concerns requires safeguards at the system, policy, and application levels, including responsible deployment practices, content moderation, and appropriate governance.

Overall, this work aims to advance the efficiency and scalability of machine learning systems. We believe that the societal impact of such advances depends primarily on how these models are deployed, rather than on the optimization techniques themselves.

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

# A. More Qualitative Result

Below in Fig. 8 and Fig. 9, we have shown more qualitative results for HilbertA and different baseline methods across various resolutions.

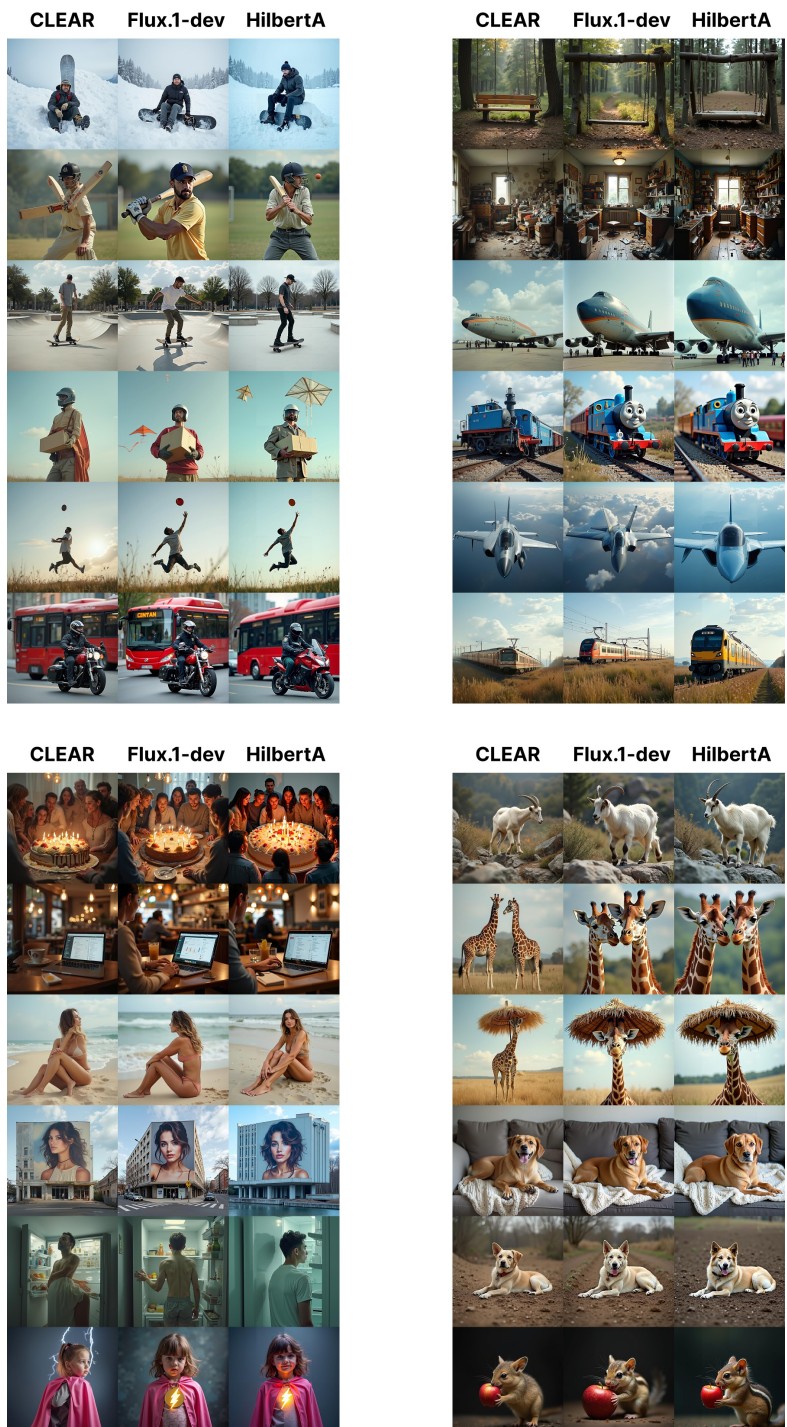

*Figure 8.* 1024 × 1024 Qualitative visual comparison between CLEAR, Flux.1-dev, and HilbertA

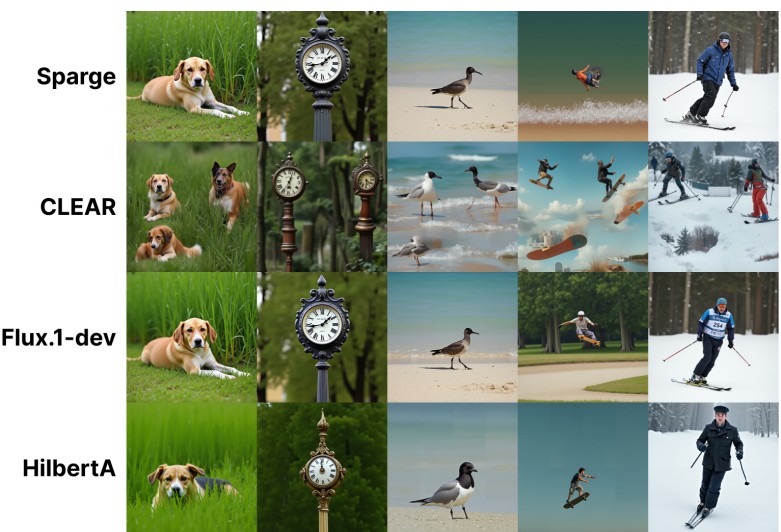

*Figure 9.* 2048 × 2048 Qualitative visual comparison between CLEAR, `Flux.1-dev`, and HilbertA

# B. Shared Region Visualization

In this section, we demonstrate the visualization of our shared region in the image hidden states.

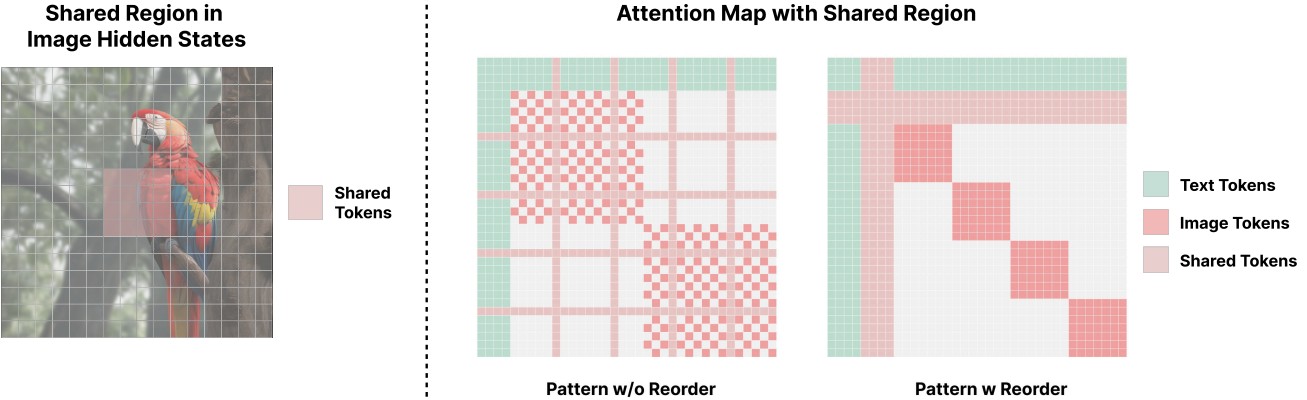

*Figure 10.* Illustration of attention map with shared region, with and without applying HilbertA reordering.

## C. Additional Ablation Results

We provide additional qualitative results for the ablation studies summarized in Table 2. The main paper reports the quantitative effects of the sliding cycle and shared region, while this appendix visualizes their qualitative impact on generation quality.

**Sliding mechanism.** Figure 11 compares HilbertA with and without the sliding mechanism under the configuration of `num_of_tiles` = 4 at $1024 \times 1024$ resolution. Without sliding, the generated images exhibit clear boundary artifacts across tiles, indicating insufficient cross-tile information exchange. Incorporating sliding substantially mitigates these artifacts by allowing information to propagate across neighboring Hilbert tiles while preserving contiguous memory access. This qualitative comparison supports the role of sliding as the main mechanism for cross-tile communication.

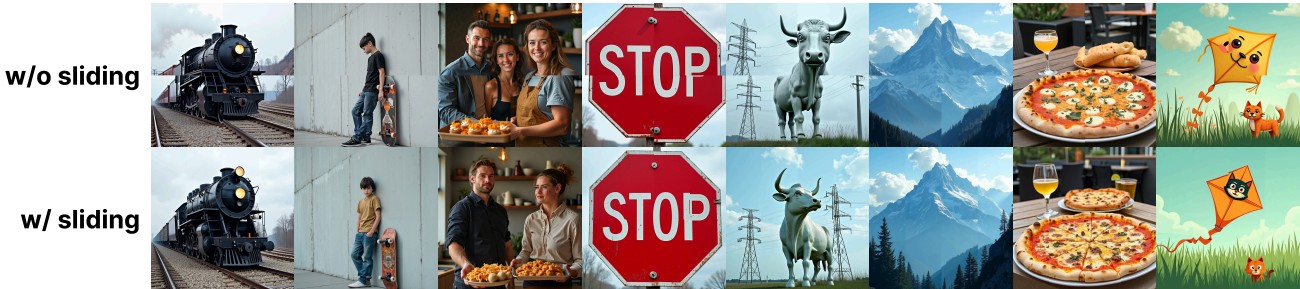

*Figure 11.* Qualitative comparison of HilbertA with and without the sliding mechanism under the configuration of `num_of_tiles` = 4 on 8 prompts at $1024 \times 1024$ resolution. **Above:** Without sliding, noticeable boundary artifacts emerge across tiles, leading to visible inconsistencies. **Below:** Incorporating sliding alleviates these artifacts by enabling effective cross-tile information exchange.

**Shared region.** Figure 12 shows qualitative comparisons using different shared-region sizes at $1024 \times 1024$ resolution. Consistent with the quantitative results in Table 2, adding a shared region improves visual coherence compared with using no shared region. A $16 \times 16$ shared region produces more coherent generations than either no sharing or a smaller $8 \times 8$ shared region, supporting our choice of $16 \times 16$ at $1024 \times 1024$ and $32 \times 32$ at $2048 \times 2048$ as the default configurations.

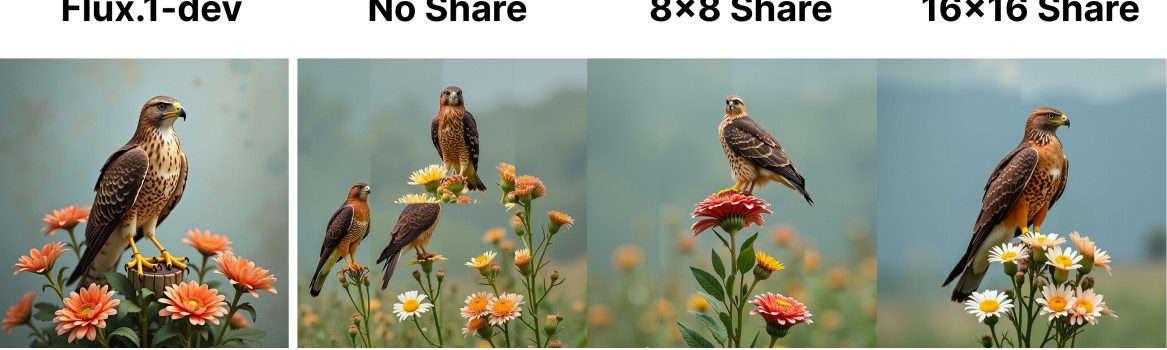

*Figure 12.* Visual comparison of image generation with different shared-region sizes at $1024 \times 1024$ resolution.

# D. Training from Scratch

We trained HilbertA from scratch using VA-VAE-f16d32 and Lightning DiT-B/1 on a 100-class subset of ImageNet-1k (140k images), on four A100 GPUs with DDP and a per-device batch size of 512 for 200k steps at a learning rate of $5 \times 10^{-4}$. The model converged to an FID of 14.10 (with 5k generated images computed against the original distribution), compared to a naive baseline of 13.08. From Fig. 13, both training runs exhibit a rapid initial decline in loss followed by gradual stabilization around 0.30, indicating stable and efficient convergence behavior of HilbertA when training from scratch.

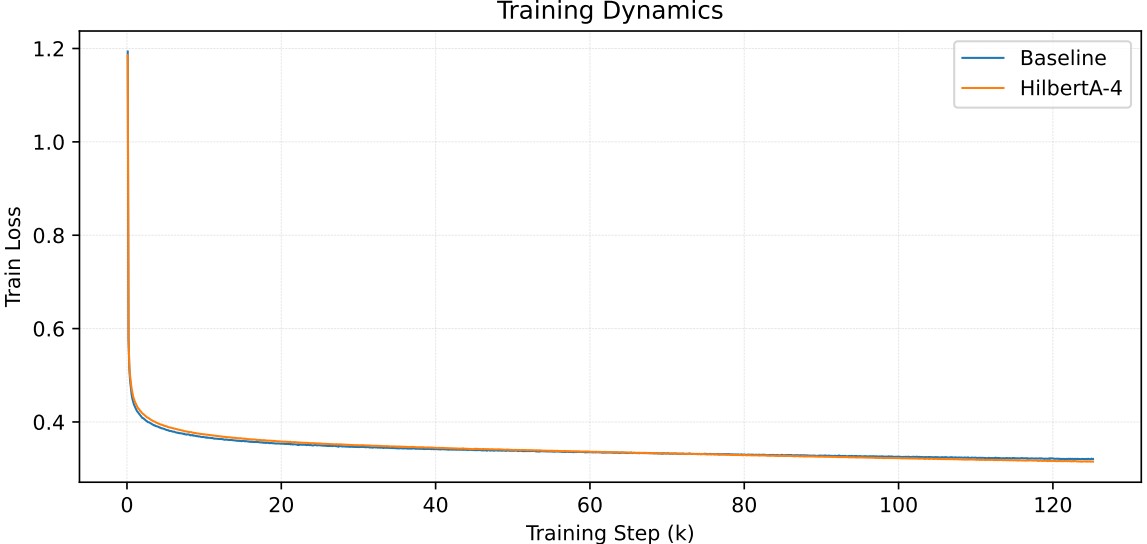

*Figure 13.* Training loss vs. training step (k) for the baseline and HilbertA-4. Across 0-120k steps, the two curves closely track each other with similar decay and stability, indicating that Hilbert-curve ordering preserves from-scratch trainability comparable to the baseline.

# E. Comparison with Additional Sparse Attention Methods

## E.1. Comparison with Swin Transformer

Although both our sparse attention mechanism and the Swin Transformer restrict attention to local neighborhoods and exchange information through interleaving sparse patterns, their formulations and system-level implications are quite different.

**HilbertA** first permutes the 2D grid into a 1D Hilbert order and then applies attention on contiguous segments along this curve. Sliding along the ordering can yield up to `tile_size` distinct sparse patterns without any intermediate memory copying, padding, or loss of contiguity. This provides fine-grained control over locality and coverage while preserving a block-aligned dense-matrix layout that is well-suited for efficient GPU execution. Moreover, the global reordering is performed only twice—once before and once after generation, and no per-layer pattern-dependent memory copies are needed. **Swin**, on the other hand, uses axis-aligned 2D windows and obtains cross-window communication by shifting the image diagonally. In practice, this shifting requires copying feature maps into a new memory layout to achieve high efficiency, so the pattern alternation would incur extra memory-copy overhead every time the pattern changes.

We next address whether the Swin rolling strategy can be directly combined with HilbertA. Swin's shifted-window scheme is defined by two distinct 2D tilings of the feature map, a standard tiling and a shifted one, each composed of fixed square windows. The framework of HilbertA, however, is built around a single curve reordering and tiles formed by contiguous 1D segments, which correspond to irregular but locality-preserving 2D regions. As long as we keep this design (one global reorder plus contiguous segments), a simple sliding operation along the curve cannot reproduce the pair of complementary square-window partitions used in Swin. Emulating Swin-style rolling would require either (i) non-contiguous index sets within each tile or (ii) different reordering across layers, both of which break the uniform, memory-contiguous layout that HilbertA explicitly targets. Thus, Swin's rolling cannot be implemented as a straightforward variant of Hilbert sliding without giving up our main efficiency advantages.

To further clarify the relationship to Swin, we implemented a Swin-like baseline by integrating the shifted-window pattern-switching mechanism into the Flux attention module, without using Hilbert reordering or sliding. This baseline mimics Swin's local-window attention and rolling behavior under identical sampling settings. As shown in Table 5, Swin does not match the generation latency or the generation quality achieved by HilbertA.

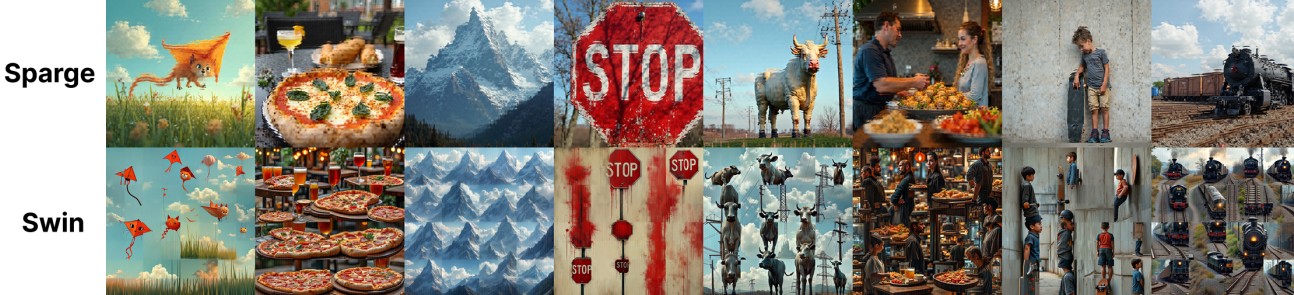

*Figure 14.* Qualitative comparison between Sparge Attention and Swin-style attention. Sparge retains overall structure but exhibits softened textures and blurred high-frequency details due to similarity-based block merging. Swin-style attention shows window-aligned seams and repeated patterns, reflecting limited cross-window communication.

Figure 14 highlights the corresponding qualitative differences. Swin-style attention tends to produce pronounced block artifacts aligned with its local windows: the pattern-switching mechanism does not propagate information as effectively across neighboring windows, leading to inconsistent global structure, multiple loosely related sub-images, and visible seams between tiles.

Taken together with the quantitative and qualitative results above, these findings demonstrate that Swin-style sparse attention is fundamentally limited in both efficiency and expressive capacity within our setting. Its pattern-switching mechanism incurs unavoidable per-switch memory-copy overhead and fails to propagate information effectively across windows, leading to suboptimal global consistency and higher FID. In contrast, HilbertA achieves lower latency and better image fidelity through a single memory-contiguous Hilbert ordering and sliding-based pattern diversity, making it a more scalable and robust sparse-attention formulation for large-scale generative modeling.

*Table 5.* Comparison of Swin-style window attention, HilbertA, and Sparge Attention.

| Method | Sparsity (%) | FID ↓ | End-to-End (s) ↓ |
|---|---|---|---|
| Swin (window_size = 16) | 93.75 | 38.52 | 14.62 |
| HilbertA (4 tiles) | 75.00 | 33.50 | 12.83 |
| HilbertA (16 tiles) | 94.00 | **31.30** | **12.57** |
| Sparge Attention | 94.95 | 32.61 | 16.36 |

### E.2. Comparison with Sparge Attention

We further tune the Sparge Attention variant to match the sparsity level of HilbertA (94.95% average sparsity) for a fair comparison. As shown in Tab. 5, Sparge Attention achieves a competitive—or even slightly better—FID score. However, its computational characteristics fundamentally limit its practical efficiency in image generation.

Sparge relies on computing block-level similarities and dynamically merging similar regions, which introduces substantial overhead when the sparsity becomes high. While this amortizes well in video diffusion models where sequence lengths are extremely large, and attention dominates runtime as reported in their paper, the overhead becomes disproportionately expensive in image generation, where the sequence length is much shorter, and attention is not the runtime bottleneck. As a result, the expected sparsity-induced speedup is negated, resulting in slower generation than HilbertA despite having comparable sparsity.

Moreover, although Sparge Attention reports a favorable FID, the generated samples present noticeable visual artifacts due to its similarity-driven block merging. As shown in Fig. 11, the method tends to oversmooth high-frequency regions, leading to blurry textures, washed-out boundaries, and loss of fine detail. For example, the fur of the cat becomes smoothed out, the pizza crust lacks crispness, the mountain ridges lose sharp definition, and the stop sign and cow exhibit diluted surface details. These high-frequency distortions are much more pronounced in images than in videos, further confirming that Sparge's adaptive grouping mechanism does not transfer well to high-resolution image generation.

This comparison highlights the importance of our design philosophy: prioritizing memory-contiguous operations and lightweight reordering enables HilbertA to deliver both high-quality images and real-time generation speed, while avoiding the computational and perceptual drawbacks associated with dynamically computed sparse patterns.

# F. More Results on HilbertA Hybridized with Full Attention

In this section, we report the effectiveness of hybrid HilbertA attention with full attention at only three denoising steps (14, 26, and 27) out of 28 total steps. This step schedule is motivated by our empirical observation that steps 11-14 primarily establish global image structure, while the final two steps focus on refinement and artifact correction.

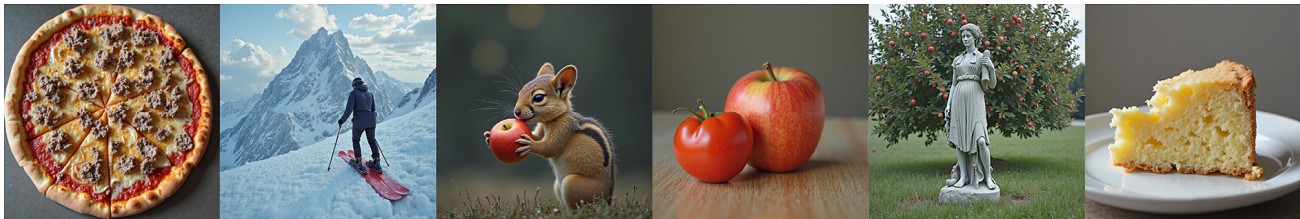

*Figure 15.* Qualitative result of hybridizing HilbertA with 3 steps of full attention out of a total of 28 steps on 1024×1024 resolution, most artifacts disappeared.

As illustrated in Fig. 15, which shows 1024×1024 samples generated using the hybrid architecture with HilbertA (4 tiles), the resulting images are clean and visually coherent, with noticeably fewer artifacts. This indicates that a small amount of full attention can be effectively incorporated into HilbertA to preserve high image quality.

| Method | FID |
|---|---|
| Flux.1-dev | 30.6 |
| HilbertA (4 tiles) | 33.5 |
| Hybrid | 30.8 |

*Table 6.* Quantitative comparison of attention strategies. The hybrid method attains near-baseline FID with minor full-attention usage.

Tab. 6 offers supporting quantitative results. Incorporating only a small number of full-attention steps already brings the FID close to the full-attention baseline, showing that HilbertA is an effective backbone and benefits noticeably from even minor full-attention supplementation.

# G. Additional VLM Baselines

We further compare HilbertA with NATTEN (Hassani et al., 2023), a 2D-aware sparse-attention baseline that restricts each token to a local $k \times k$ neighborhood. NATTEN preserves local 2D structure, but its neighborhoods do not generally correspond to contiguous memory segments under the standard flattened token layout. It therefore provides a useful comparison for separating spatial locality from hardware-aligned memory access. We report attention-kernel latency and benchmark performance on `Qwen3-VL-8B`.

| Tokens | Flash | HilbertA | NATTEN-7 | NATTEN-21 |
|--------|-------|----------|----------|-----------|
| 1,024  | 0.10  | 0.31     | 0.22     | 1.06      |
| 4,096  | 0.64  | 0.56     | 0.35     | 3.43      |
| 9,216  | 2.97  | 2.16     | 0.65     | 7.33      |
| 16,384 | 8.99  | 4.47     | 1.06     | 12.39     |
| 30,976 | 31.93 | 16.12    | 1.89     | 24.07     |

*Table 7.* Attention-kernel latency in milliseconds on `Qwen3-VL-8B` across sequence lengths. For each token length, the  best  and  2nd best  values are highlighted. HilbertA scales substantially better than FlashAttention and NATTEN-21 at long sequence lengths, achieving a $2.0\times$ kernel speedup over FlashAttention and a $1.5\times$ speedup over NATTEN-21 at 30,976 tokens. Although NATTEN-7 achieves the lowest latency at medium and long sequence lengths, Table 8 shows that this comes with a severe quality drop.

As shown in Table 7, HilbertA provides a strong efficiency advantage in the long-context regime. At 30,976 tokens, HilbertA reduces kernel latency from 31.93 ms to 16.12 ms compared with FlashAttention, and from 24.07 ms to 16.12 ms compared with NATTEN-21. NATTEN-7 obtains the lowest kernel latency because it uses a much smaller local neighborhood, but this aggressive sparsity substantially degrades model quality, as shown below.

| Method | OCRBench $\uparrow$ | MMStar $\uparrow$ | MMBench $\uparrow$ |
|--------|---------|--------|---------|
| HilbertA   | 686 | 44.6% | 0.705 |
| NATTEN-7   | 426 | 37.1% | 0.599 |
| NATTEN-21  | 667 | 44.2% | 0.725 |

*Table 8.* Benchmark performance on `Qwen3-VL-8B` across different sparse-attention methods. The  best  and  2nd best  values are highlighted for each metric. HilbertA achieves the best OCRBench and MMStar scores while maintaining competitive MMBench performance. NATTEN-7 is fast but substantially degrades model quality, whereas increasing the NATTEN neighborhood size recovers part of the performance.

The quality results in Table 8 show that the fastest NATTEN configuration is not a favorable quality–efficiency trade-off: NATTEN-7 drops substantially on OCRBench, MMStar, and MMBench. Using a larger NATTEN neighborhood recovers part of the performance, but the resulting method no longer offers the same aggressive-efficiency regime as NATTEN-7. In contrast, HilbertA maintains strong performance across benchmarks while still achieving practical kernel speedups at long sequence lengths. These results support the central design choice of HilbertA: for 2D structured data, effective sparse attention should preserve spatial locality while also aligning the sparse layout with contiguous GPU memory access.

# H. Additional Analysis of Scanning Curves

## H.1. Practical Differences Among Scanning Curves

Although many one-dimensional scan orders can be implemented with contiguous memory layouts, their ability to preserve two-dimensional neighborhoods differs substantially in practice. In this section, we provide an empirical and qualitative comparison of commonly used scanning curves, including serpentine, Morton (Z-order), and Hilbert curves, to clarify their practical implications for sparse attention on 2D image data.

We emphasize that the key property of interest is not memory contiguity alone, but how well a one-dimensional ordering preserves local two-dimensional neighborhoods under sparse attention. For example, serpentine traversals maintain horizontal adjacency but systematically separate vertical neighbors by an entire row. As a result, when local attention is applied, vertical neighborhoods are consistently broken across window boundaries. Morton curves improve hierarchical coherence by interleaving bits across dimensions, but tend to fragment local neighborhoods at fine spatial scales.

By contrast, Hilbert curves preserve horizontal and vertical adjacency more isotropically, owing to their recursive and self-similar construction. This property is reflected quantitatively by the proposed metrics, where Hilbert curves consistently achieve lower geometric distortion error (GDE) and edge average stretch compared to alternative scan orders across grid sizes.

## H.2. Empirical Validation on Diffusion Models

To further validate that these differences translate into practical performance gaps, we conduct an additional experiment on `LightningDiT-XL`. We start from the released 800-epoch pretrained checkpoint and finetune the model for 100k steps on a 100-class ImageNet subset, using a fixed learning rate of $1 \times 10^{-4}$ and a batch size of 64 per GPU ($4 \times$ RTX 6000 Ada, DDP). We then evaluate generation quality using gFID on 50k classifier-free guided samples drawn from the same 100 classes.

*Table 9.* Ablation over different scanning curves on `LightningDiT-XL`. Lower is better for both gFID and GDE.

| Scan Order | gFID ↓ | GDE ↓ |
|---|---|---|
| Full Attention | 10.383 | – |
| Hilbert Curve | **10.442** | **2.965** |
| Morton Curve | 10.501 | 4.747 |
| Serpentine Curve | 10.552 | 4.397 |

Table 9 summarizes the results. Among the evaluated scan orders, Hilbert curves consistently achieve the best gFID, followed by Morton and then serpentine orderings. This ranking closely mirrors the ordering induced by GDE, indicating that the proposed metric correlates with downstream generation quality. We note that the absolute performance gaps are modest in this setting, as `LightningDiT-XL` operates on a $16 \times 16$ latent grid, where GDE differences across scan orders are also limited. As the latent resolution increases, GDE diverges more sharply across curves, suggesting that the corresponding performance gaps are likely to widen at higher resolutions.

# I. Consumer-Grade GPU Efficiency

To further evaluate whether the efficiency gains of HilbertA transfer beyond datacenter GPUs, we benchmark the attention kernels on an NVIDIA RTX 4090. This setting provides a complementary test of hardware generalization on a consumer-grade GPU with a different memory hierarchy and a smaller memory budget. Since the RTX 4090 has 24GB of VRAM, full `Flux.1-dev` BF16 end-to-end inference is not feasible at the target setting. We therefore report attention-kernel latency for Flux.1, and additionally report both attention-kernel latency and end-to-end latency on the lighter `FLUX.2-Klein-9B` model using 10 denoising steps at 1K resolution. We do not include Sparge Attention in this comparison, as its current implementation does not support the RTX 5090 architecture used in our consumer-GPU testing environment.

*Table 10.* Consumer-grade GPU efficiency on an NVIDIA RTX 4090. We report attention-kernel latency for `Flux.1-dev`, and both attention-kernel latency and end-to-end latency for `FLUX.2-Klein-9B`. HilbertA consistently achieves the lowest latency across settings, despite using lower sparsity than CLEAR-r8 in some cases.

| Method | Sparsity | `Flux.1` Attn. (ms)↓ | `Flux.2` Attn. (ms)↓ | `Flux.2` E2E (s)↓ |
|---|---|---|---|---|
| Baseline | 0% | 1.67 | 2.33 | 5.84 |
| HilbertA 4tile | 75% | 1.14 | 1.41 | 5.58 |
| HilbertA 16tile | 94% | **0.78** | **0.92** | **5.17** |
| CLEAR-r8 | 99% | 1.19 | 1.30 | 5.52 |
| CLEAR-r16 | 80% | 1.21 | 1.67 | 5.65 |

As shown in Table 10, HilbertA achieves the fastest latency across all tested settings on the RTX 4090. On Flux.1, HilbertA with 16 tiles reduces attention-kernel latency from 1.67 ms to 0.78 ms, corresponding to a 53.3% reduction. On FLUX.2, the same configuration reduces attention latency from 2.33 ms to 0.92 ms and improves end-to-end latency from 5.84 s to 5.17 s, yielding an 11.5% end-to-end reduction.

Notably, HilbertA also outperforms CLEAR-r8 even though CLEAR-r8 uses higher sparsity, 99% versus 94% for HilbertA 16tile. This result further supports our central observation that sparsity alone does not determine practical speedup. Instead, the memory access pattern plays a critical role: by preserving contiguous access after Hilbert-curve reordering, HilbertA allows sparse computation to translate more directly into wall-clock acceleration. These results indicate that the layout-aware design of HilbertA is effective not only on datacenter GPUs, but also on consumer-grade hardware.

## J. Training Budget Comparison

In addition to the quantitative comparison in the main paper, we further examine the effect of training budget on sparse-attention adaptation. HilbertA is adapted with lightweight LoRA fine-tuning using 10k images for four epochs, while CLEAR typically relies on a substantially larger distillation setup with 75 epochs and updates a much larger fraction of model parameters. To isolate the role of training budget, we also train CLEAR under the same data budget used for HilbertA. Under this limited-data setting, CLEAR exhibits noticeably degraded visual quality, indicating that its adaptation procedure requires substantially more data to recover generation fidelity. In contrast, HilbertA maintains a favorable quality–efficiency trade-off under the same lightweight fine-tuning regime, suggesting that the Hilbert-ordered locality prior provides a stronger inductive bias for sparse attention in diffusion models.

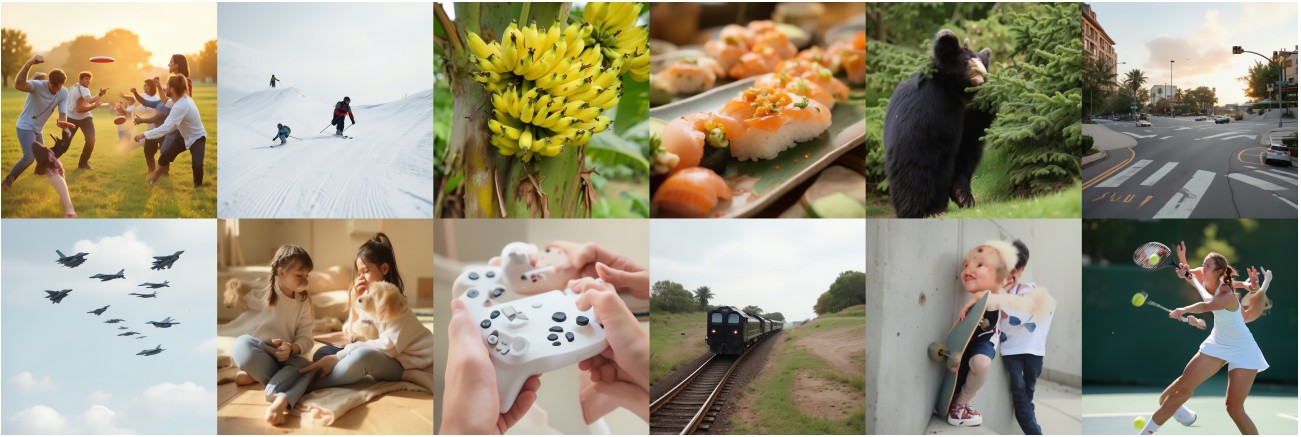

*Figure 16.* Qualitative results of CLEAR under the same lightweight training-data budget used for HilbertA. CLEAR is trained using the same 10k-image, four-epoch budget as HilbertA, rather than its standard large-scale distillation setup. Under this limited-data setting, CLEAR produces noticeably degraded samples, suggesting that it requires substantially more training data to recover image quality. This comparison highlights the practical advantage of HilbertA, whose Hilbert-ordered locality prior enables effective sparse-attention adaptation with a much smaller training budget.

# K. General Hilbert Curve Algorithm

**Pseudocode.** For completeness, we provide a high-level pseudocode specification of the generalized Hilbert construction used in HilbertA. The algorithm follows the recursive construction of Červený (2021), which extends the standard Hilbert curve to arbitrary rectangular grids. At each step, the current rectangle is decomposed into smaller sub-rectangles while preserving a continuous traversal path. For highly elongated rectangles, the algorithm performs a two-way split along the long axis; otherwise, it applies a Hilbert-like three-way decomposition. When both dimensions are powers of two, this construction reduces to the standard Hilbert curve.

---

**Algorithm 1** Generalized 2D Hilbert Curve Construction

---

**Require:** Grid width $W$ and height $H$
**Ensure:** A continuous ordering of all $W \times H$ grid cells
 1: **function** GENERALHILBERT2D($W, H$)
 2:     **if** $W \geq H$ **then**
 3:         **return** FILL$(0, 0, (W, 0), (0, H))$
 4:     **else**
 5:         **return** FILL$(0, 0, (0, H), (W, 0))$
 6:     **end if**
 7: **end function**
 8: **function** FILL$(x, y, \mathbf{a}, \mathbf{b})$                     ▷ $\mathbf{a}$: major direction, $\mathbf{b}$: minor direction
 9:     **if** $\|\mathbf{a}\| = 1$ **or** $\|\mathbf{b}\| = 1$ **then**
10:         **return** traverse the remaining row or column in order
11:     **end if**
12:     $\mathbf{a}_1 \leftarrow \lfloor \mathbf{a}/2 \rfloor, \quad \mathbf{a}_2 \leftarrow \mathbf{a} - \mathbf{a}_1$
13:     $\mathbf{b}_1 \leftarrow \lfloor \mathbf{b}/2 \rfloor, \quad \mathbf{b}_2 \leftarrow \mathbf{b} - \mathbf{b}_1$
14:     **if** the rectangle is highly elongated along $\mathbf{a}$ **then**             ▷ Two-way split
15:         **if** $\|\mathbf{a}_1\|$ is odd **and** $\|\mathbf{a}\| > 2$ **then**
16:             adjust $\mathbf{a}_1$ by one unit to make the split well-formed
17:             $\mathbf{a}_2 \leftarrow \mathbf{a} - \mathbf{a}_1$
18:         **end if**
19:         FILL$(x, y, \mathbf{a}_1, \mathbf{b})$
20:         FILL$(x + a_{1x}, y + a_{1y}, \mathbf{a}_2, \mathbf{b})$
21:     **else**                                    ▷ Hilbert-like three-way split
22:         **if** $\|\mathbf{b}_1\|$ is odd **and** $\|\mathbf{b}\| > 2$ **then**
23:             adjust $\mathbf{b}_1$ by one unit to make the split well-formed
24:             $\mathbf{b}_2 \leftarrow \mathbf{b} - \mathbf{b}_1$
25:         **end if**
26:         FILL$(x, y, \mathbf{b}_1, \mathbf{a}_1)$
27:         FILL$(x + b_{1x}, y + b_{1y}, \mathbf{a}, \mathbf{b}_2)$
28:         FILL$(x + (a_x - b_{1x}), y + (a_y - b_{1y}), -\mathbf{b}_1, -\mathbf{a}_2)$
29:     **end if**
30: **end function**

---

