# OpenReview forum: "HilbertA: Hilbert-Curve–Aligned Sparse Attention for 2D Structured Data"
_ICML.cc/2026/Conference — ICML 2026 regular_

### Official Review · Reviewer_NtrY · 2026-03-11

**Soundness:** 2
**Presentation:** 3
**Significance:** 2
**Originality:** 3
**Overall Recommendation:** 4
**Confidence:** 4

**Summary:**

This paper proposes using the Hilbert Curve to map 2D image tokens into a 1D sequence, which both preserves spatial locality and achieves contiguous GPU memory access. By implementing a customized kernel in Triton, the method demonstrates significant speedup when processing high-resolution images such as 2048x2048.

**Compliance With Llm Reviewing Policy:**

Affirmed.

**Final Justification:**

Concerns addressed.

**Key Questions For Authors:**

1. Conduct experiments on other recent SOTA models, including Qwen-image, FLUX.2, etc.

2. It is necessary to prove that the decline in visual quality is not as obvious or unacceptable as it currently appears. Testing on more models might help demonstrate the effectiveness and validity of the method.

**Limitations:**

Please see the Weakness.

**Strengths And Weaknesses:**

Strengths:
1. The authors propose a novel sparsification method specifically for generative models.

2. The paper provides a comparative analysis of GDE across different scanning orders such as Morton. It effectively demonstrates the superiority of the Hilbert curve.

Weaknesses:
1. The visual results are unsatisfactory. Even in the images provided in the paper, there are severe blocking effects and trajectory artifacts. For example, the third image of HilbertA in Figure 8.

2. The degradation in FID and other visual metrics is too pronounced, which aligns with the qualitative visual results and subjective perception. In my view, this loss of quality is excessive and unacceptable.

3. The authors claim that fine-tuning is required. Thus, they need to conduct tests on a wider variety of models rather than just Flux.1-dev.

4. Although there is an advantage at 2K resolution, at 1024 resolution, the benefits of HilbertA are not significant compared to existing methods like CLEAR. Most daily usage remains around the 1K resolution range.

---

> ### Author Rebuttal · Authors · 2026-03-31
>
> We thank the reviewer for the thoughtful questions. Below, we address these points.
>
> ## W1,2 & Q2. Visual Artifacts
>
> We thank the reviewers for raising this concern. We address it along three dimensions: additional qualitative evidence, contextualization on metrics, and a practical mitigation strategy.
>
> ### 1.1 More Qualitative
>
> We agree that tile-boundary artifacts can occasionally appear, as discussed in Section 4.2. However, these artifacts are localized to tile boundaries and occur in a small fraction of generated images—the majority of samples (Figures 5, 7) are artifact-free. To provide a more comprehensive assessment, we added two sets of results, neither of which is cherry-picked.
>
> - [50 additional COCO-prompted samples](https://anonymous.4open.science/r/hilberta/1_coco50_vs_flux_grid.md);
> - [200 real-world user-prompted samples](https://anonymous.4open.science/r/hilberta/2_t2iprompt200.md) drawn from t2i-prompts-3m (Chen et al., 2025).
>
> ### 1.2 Contextualizing FID
>
> We note that the FID numbers require context. At 1K, HilbertA (16 tiles) achieves FID 31.3, outperforming all CLEAR configs (32.2–33.0). At 2K, the 4-tile (FID 38.4) is competitive with CLEAR r=16 (39.3) and better than CLEAR r=8 (43.2). The higher FID of the 16-tile configuration (47.5) reflects an explicit quality–efficiency trade-off at maximum sparsity.
>
> We further note that this degradation is not consistently observed across models. On *FLUX.2* (section below), HilbertA maintains comparable or improved quality while achieving efficiency gains, suggesting that the observed FID drop in *Flux.1-dev* is **not an inherent limitation** of our approach.
>
> ### 1.3 Mitigation
>
> Even on *FLUX.1*, the gap is easily closed: inserting just 3 full-attention steps out of 28 reduces FID from 33.5 to 30.8, nearly matching the baseline (30.6), while eliminating most visible artifacts. (Appendix G, Figure 14). This demonstrates that the quality gap is small and easily recoverable with minimal overhead.
>
> ## W3 & Q1. FLUX.2 Results
>
> We additionally evaluate HilbertA on *FLUX.2-Klein-9B*, fine-tuning on only 40k images (vs. 750k for CLEAR). We report attention kernel latency for each method, as well as end-to-end generation time with 10 denoising steps.
>
> ||FID|LPIPS|CLIP|Attn (ms)|E2E (s)|
> |-|-|-|-|-|-|
> |FLUX.2-Klein|26.39|100|100|1.72|4.46|
> |HilbertA 4tile|26.26|**51.6**|**90.3**|1.20|4.32|
> |HilbertA 16tile|**25.67**|60.0|85.1|1.16|4.17|
> |Sparge (71%)|28.03|64.5|82.7|1.39|4.35|
> |CLEAR-r8 (69.5%)|31.59|61.5|84.5|1.19|4.28|
> |CLEAR-r16|29.77|64.7|83.2|1.54|4.40|
>
> HilbertA (4 tiles) improves FID from 26.39 to 26.26 while reducing attention latency by 30% (1.72→1.20 ms) and end-to-end generation time from 4.46s to 4.32s. The 16-tile configuration pushes efficiency further, achieving the best FID (25.67) and the fastest end-to-end time (4.17s).
>
> Among competing methods, Sparge Attention shows worse quality (28.03 FID) with slower attention (1.39 ms); CLEAR-r8 achieves a comparable kernel time (1.19 ms) but suffers a substantial quality drop (31.59 FID), and CLEAR-r16 is both slower (1.54 ms) and lower quality (29.77 FID) - both despite being trained on 18× more data. [Qualitative results here](https://anonymous.4open.science/r/hilberta/3_flux2_comparison.md).
>
> These results demonstrate that HilbertA generalizes effectively across model architectures, achieving the best overall quality–efficiency tradeoff at significantly lower tuning cost.
>
> ## W4 Advantage at 1k resolution vs. CLEAR
>
> At 1K, HilbertA already demonstrates a favorable quality–efficiency tradeoff: the 4-tile variant is faster than all CLEAR configurations at comparable quality, while the 16-tile achieves the best FID (31.3) and the fastest runtime.
>
> On *FLUX.2-Klein-9B*, the advantage is even clearer. Both HilbertA configurations achieve comparable FID to the dense baseline (26.26 and 25.67 vs. 26.39), and much lower than CLEAR (~31), confirming that the benefit is not limited to high-res and models.
>
> Another key practical distinction is training cost. HilbertA uses lightweight LoRA fine-tuning (~1% of parameters, and only 40k images), whereas CLEAR relies on large-scale distillation (\~70% of parameters, 750k images). Despite this 18× gap in data, CLEAR underperforms HilbertA on *FLUX.2* across all metrics. This suggests that the Hilbert-ordered locality provides a strong inductive bias that reduces the need for extensive fine-tuning, making it the more practical solution.
>
> We also note that the trend toward high-res is clear. Modern models increasingly operate at 2K+ for applications such as print media. Even in current workflows, AIGC practitioners typically generate at 1k and then upsample, making high-res efficiency directly relevant. In this regime, HilbertA's advantage becomes substantially more pronounced.
>
> ---
> References
>
> [1] Chen, Hansheng, et al. "pi-flow: Policy-based few-step generation via imitation distillation." arXiv preprint arXiv:2510.14974 (2025).

---

> > ### Author Rebuttal · Reviewer_NtrY · 2026-04-02
> >
> > The visualizations are satisfactory. However, why can LPIPS exceed 1?

---

> > > ### Author Response · Authors · 2026-04-02
> > >
> > > We sincerely thank the reviewer for the positive feedback on our visual quality and for pointing out the issue regarding LPIPS reporting.
> > >
> > > To clarify, we computed LPIPS following the standard protocol. The raw LPIPS values fall within a typical range (e.g., around 0–1 in our case) for our dataset and experimental setup. In the paper, we scaled the values by ×100 for ease of presentation.
> > >
> > > The metric itself is computed correctly, but the presentation may be misleading. We will revise in our next version to either report the raw values directly or explicitly indicate the ×100 scaling in the table caption.
> > >
> > > For reference, the actual (unscaled) LPIPS values are as follows:
> > >
> > > |  | LPIPS |
> > > | --- | --- |
> > > | FLUX.2 Klein 9B | - |
> > > | HilbertA Lora 4 tile | **0.516** |
> > > | HilbertA Lora 16 tile | 0.600 |
> > > | Sparge (71% Avg Sparsity) | 0.645 |
> > > | CLEAR-r8 | 0.615 |
> > > | CLEAR-r16 | 0.647 |

---

### Official Review · Reviewer_D17g · 2026-03-13

**Soundness:** 3
**Presentation:** 3
**Significance:** 3
**Originality:** 3
**Overall Recommendation:** 5
**Confidence:** 4

**Summary:**

This paper proposes HilbertA, a sparse attention mechanism designed for 2D structured data such as images in diffusion models and vision-language models. The key idea is to reorder image tokens along a Hilbert space-filling curve, which preserves spatial locality while producing a contiguous memory layout that aligns well with GPU execution. They propose the sliding mechanism and shared tile that enables cross-tile communication. Experiments on image diffusion models and vision language models show notable speedups while maintaining comparable model performance.

**Compliance With Llm Reviewing Policy:**

Affirmed.

**Final Justification:**

The rebuttal is solid so I keep my score of 5.

**Key Questions For Authors:**

N/A

**Strengths And Weaknesses:**

Strengths:
1. The use of Hilbert curves to improve memory contiguity is intuitive and well-aligned with GPU execution characteristics. Though it is applied in SpargeAttention, the authors proposed some new features that are useful to improve the quality.
2. The idea of a sliding mechanism and a shared tile that enables cross-tile communication is very novel. The effectiveness is also being validated through ablation studies.
3. The paper is well-written and easy to understand. The figures are clear and precise.

Weaknesses:
Though they are not critical flaws, I think by addressing the following comments can improve the paper:
1. How does the Hilbert curve appear when the sequence length is not an exponent of 2? Though discussed in Appendix C, I suggest that authors give a formal definition / pseudo code about how it is being constructed, and what the curve looks like when N is a relatively large number to help readers understand it (For example, N = 36 / 42 / ...)

2. Ablation studies in Appendix D should be moved to the main text. They are very important parts to validate the effectiveness of the proposed method.

3. SpargeAttention - 17% sparsity is strange. The authors should double-check the result.

4. (Optional to include in rebuttal) The paper would be more solid if more results and comparision with baselines(SVG, STA, etc.) on video diffusion models can be displayed.

---

> ### Author Rebuttal · Authors · 2026-03-31
>
> We thank the reviewer for the thoughtful suggestion and kind appreciation of our work.
>
> ## 1. Non-power-of-2 Hilbert Curve Construction
>
> We thank the reviewer for this suggestion. To extend the classical Hilbert curve to arbitrary-size rectangles, we adopt a generalized Hilbert curve construction proposed by Červený (2021).
>
> The key idea is straightforward: instead of requiring power-of-two dimensions, the algorithm recursively splits any rectangle into sub-rectangles using the same U-shaped template as the standard Hilbert curve.
>
> For flat rectangles, it uses a simpler two-way split along the long axis; otherwise (i.e., for roughly square shapes), it applies a three-way decomposition (up → across → down). When halving a dimension yields an odd number, it rounds up to the next even number to keep the partition well-formed.
>
> This adjustment guarantees a continuous path without diagonal jumps. When both dimensions are powers of two, the algorithm reduces exactly to the standard Hilbert curve, so all our existing results remain unchanged. We will include the pseudocode (shown below) and a worked example in the revised appendix.
>
> We kindly refer the reviewer to a visualization of this pattern in our [anonymous GitHub notebook](https://anonymous.4open.science/r/hilberta/4_hilbert_layout_nonpower2.md).
> Please let us know if there are any issues accessing the link. These examples correspond to sequence lengths $N$ = 8, 16, 32, 24, and 36, demonstrating that HilbertA’s reordering generalizes seamlessly to arbitrary feature map shapes without padding or loss of contiguity.
>
> We also include a pseudocode specification of the algorithm below for clarity.
>
> ```python
> function GENERAL_HILBERT_2D(width, height):
>     if width >= height:
>         return FILL(x=0, y=0, major=(width, 0), minor=(0, height))
>     else:
>         return FILL(x=0, y=0, major=(0, height), minor=(width, 0))
>
> function FILL(x, y, major, minor):
>     if length(minor) == 1 or length(major) == 1:
>         traverse one row along major or one column along minor
>
>     half_major, half_minor = floor(major / 2), floor(minor / 2)
>
>     if rectangle is very long in the major direction:
>         # Two-way split along the major axis
>         if length(half_major) is odd and length(major) > 2:
>             adjust half_major by one unit
>         recurse on first half
>         recurse on second half
>     else:
>         # Standard Hilbert-like three-way split: up → across → down
>         if length(half_minor) is odd and length(minor) > 2:
>             adjust half_minor by one unit
>         recurse on first small block
>         recurse on central long block
>         recurse on last small block in reversed orientation
> ```
>
> ## 2. Appendix D
>
> We thank the reviewer for the suggestion. We agree that these ablations are important for validating our method, and we will move them from Appendix D into the main text in the revised version.
>
> ## 3. Sparge Attention Sparsity Clarification (with additional results on *FLUX.2-Klein*)
>
> We thank the reviewer for raising this point. We clarify that SpargeAttention does not use a fixed or learned sparsity level. Instead, the resulting sparsity is input-dependent and governed by hyperparameters, which are tuned for the specific model and architecture.
>
> Because the official Sparge Attention repository does not provide configurations for *FLUX.1-dev* (or *FLUX.2-Klein-9B*), we conducted an extensive hyperparameter search as suggested (approximately 11.5 hours), to obtain the specs and results reported in the main table in Section 3.2. We acknowledge this is lower than typical, but it reflects the configuration that Sparge Attention's own tuning procedure converges to on this architecture.
>
> For completeness, we additionally report a higher-sparsity variant (~95%) in Appendix F.2. Also, in response to Reviewer NtrY's request for evaluation on additional models, we have tuned a configuration with ~71% average sparsity on *FLUX.2-Klein-9B*. This should be a more typical level that may better address the reviewer's concern about the anomalous 17% sparsity. We will clarify the input-dependent nature of SpargeAttention's sparsity more explicitly in the revised paper.
>
> ---
> References
>
> [1] Jakub Červený. Generalized Hilbert (“gilbert”) space-filling curve for rectangular domains of arbitrary sizes. GitHub repository, 2021.

---

> > ### Author Rebuttal · Reviewer_D17g · 2026-04-03
> >
> > I keep my score (5) to reflect my positive attitude toward this paper.

---

### Official Review · Reviewer_Sqja · 2026-03-14

**Soundness:** 3
**Presentation:** 3
**Significance:** 2
**Originality:** 3
**Overall Recommendation:** 4
**Confidence:** 4

**Summary:**

The paper proposes HilbertA, a hardware-efficient sparse attention mechanism specifically designed for 2D structured data (e.g., images). The core motivation is to resolve the conflict between maintaining spatial locality and achieving high hardware execution efficiency on GPUs. By reordering 2D tokens using a Hilbert Curve, the authors ensure that tokens adjacent in 2D space remain contiguous in 1D memory, enabling coalesced memory access. Combined with a tiling strategy and a sliding window schedule to expand the receptive field, HilbertA achieves significant speedups in high-resolution diffusion models (like Flux.1) and Vision-Language Models (like Qwen2-VL/Qwen3-VL) with minimal performance degradation.

**Compliance With Llm Reviewing Policy:**

Affirmed.

**Key Questions For Authors:**

Please see the weaknesses.

**Limitations:**

yes

**Strengths And Weaknesses:**

### Strengths

1. The paper is well writing and easy to follow.
2. The paper correctly identifies that the bottleneck of current 2D sparse attention (e.g., Swin, CLEAR) is the "uncoalesced memory access" caused by non-contiguous memory layouts. The choice of the Hilbert Curve as an inductive bias is mathematically elegant. Its fractal nature and self-similarity perfectly align with the hierarchical nature of vision transformers and tiled attention.

2. The introduction of EAS and GDE provides a rigorous quantitative framework to compare different scanning orders (Morton, Serpentine, etc.), moving beyond purely empirical observations.

3. The empirical results are impressive, particularly the 4.17x attention speedup and 1.51x end-to-end speedup for 2048 x 2048 resolution in Flux.1. It addresses the "high resolution" bottleneck in modern AIGC applications.
4. The authors validate the method across both generative and discriminative/understanding tasks. The ablation studies on scanning curves and the analysis of the reordering overhead demonstrate a comprehensive understanding of the system's trade-offs.

### Weaknesses

1. While the application to sparse attention is well-executed, using Hilbert curves for image scanning/serialization is a classical technique in image processing. The paper could better differentiate its technical contribution from prior works like Space-Filling Curves for Computer Graphics or recent sparse kernels like *SageAttention/SparseAttention* to highlight the specific architectural innovations in the attention kernel.
2. The experiments are primarily conducted on high-end NVIDIA GPUs (A100/H100). Since the paper claims "hardware-aligned" efficiency, it would be beneficial to see results on consumer-grade GPUs (RTX 4090/3090) or specialized AI accelerators to prove that the memory coalescing benefits translate across different memory architectures.
3. The performance depends on the Tile Size ($N_T$) and the Sliding Schedule ($L$). The paper lacks a systematic "rule of thumb" or sensitivity analysis on how these parameters should scale with sequence length or model depth to ensure a global receptive field without losing the efficiency gain.
4. For the VLM section (e.g., Qwen), while it shows minimal loss, comparing against other competitive sparse attention methods (like *FlashAttention-3* or *Ring Attention* adapted for 2D) would strengthen the claim that HilbertA is the superior choice for 2D data specifically.
5. In Table 2, some FID scores for HilbertA are slightly higher than the baseline. While the authors claim the difference is "negligible," a qualitative analysis (visualizing samples) to check for potential blocky artifacts or loss of global coherence due to the tiling/sliding window would be valuable.

---

> ### Author Rebuttal · Authors · 2026-03-31
>
> We thank the reviewer for the questions.
>
> ## W1 Novelty wrt Space-filling Curves & Sparse Kernels
>
> We agree that Hilbert curves are classical in image processing. Our novelty lies not in the curve itself, but in resolving the fundamental dilemma between 2D spatial locality and contiguous GPU memory access in sparse attention.
>
> The closest prior work is Sparge Attention, which also applies Hilbert curves to attention but only as a preprocessing step to identify and compress self-similar blocks. So, sparsity is determined by similarity thresholds, leading to uncoalesced memory access. HilbertA, instead, **builds the entire sparse attention around the curves**: reordering, tiling, sliding, and computation all operate on a single contiguous memory layout. This is why it achieves larger speedups even at lower sparsity.
>
> We further note that kernel-level optimizations (e.g., SageAttention, FlashAttention) are orthogonal and complementary. For instance, HilbertA's Triton kernel is already built on a FA2-style tiled design.
>
> ## W2 Consumer-grade GPUs Efficiency
>
> We thank the reviewer for highlighting hardware generalization. We conducted additional experiments on an RTX 4090 (Sparge Attention is not yet supported on the 5090 architecture). Due to its 24GB VRAM, full *Flux.1-dev* BF16 runs were infeasible, so we report kernel latency and an extra end-to-end latency on the lighter *Flux.2-Klein* (10 denoise steps, 1K res).
>
> ||Sparsity|*Flux.1* Attn (ms)|*Flux.2* Attn (ms)|*Flux.2* E2E (s)|
> |-|-|-|-|-|
> |Baseline|0%|1.67|2.33|5.84|
> |HilbertA 4tile|75%|1.14|1.41|5.58|
> |HilbertA 16tile|94%|**0.78**|**0.92**|**5.17**|
> |CLEAR-r8|99%|1.19|1.30|5.52|
> |CLEAR-r16|80%|1.21|1.67|5.65|
>
> HilbertA consistently achieves fastest across all settings, reducing *Flux.1* kernel latency by 53% and *FLUX.2-Klein* E2E by 8%. Notably, **HilbertA outperforms CLEAR even when it operates at higher sparsity** (99% vs. 94%), confirming that not sparsity alone, but memory-contiguous access drives real speedups. These results confirm HilbertA's layout-aware design translates effectively to consumer GPUs with different memory hierarchies.
>
> ## W3 Tile Size & Sliding
>
> We thank the reviewer for this inspiring question. To derive a principled guideline for how tile size and sliding schedule should scale with resolution, we extend the ERF analysis (Section 2.2) by relaxing the **perfect-transfer assumption**. A messenger token compresses an entire tile into $d$ dimensions, which is lossy. Modeling the single-hop retention rate as $\gamma \in (0,1]$ (decreasing with tile size), information arriving via $k$ hops retains a fraction $\gamma^k$. Summing over all $T$ tiles, the effective coverage becomes $\widetilde{C}\infty = \frac{1}{T}\sum{k=0}^{T-1}\gamma^k = \frac{1}{T}\cdot\frac{1-\gamma^T}{1-\gamma}$.
>
> This reveals two scaling strategies: fixing $T$ (tile count) gives a constant lower bound $\widetilde{C}\infty \geq 1/T$; fixing $N_T$ (tile size) lets $T$ grow linearly, causing $\widetilde{C}\infty \propto 1/N \to 0$. Fixing $T$ is strictly preferable.
>
> This explains why 4 tiles outperforms 16 in quality (shorter chain), why sliding cycle $L$ is insensitive ($L$ does not affect $T$ or $\gamma$), and why the same $T$ works across resolutions.
>
> We summarize a general guideline: fix $T$, not $N_T$. As resolution grows and $\gamma$ decreases, increasing $T$ is natural. $L$ can be freely chosen within $L \leq T$. We will include the full derivation in the revision.
>
> ## W4 More Baselines on VLM
>
> We clarify that FA3 and Ring Attention aren't sparse attention methods. FA3 optimizes exact attention I/O without a sparse pattern, while the latter partitions seqs across devices for distributed inference. Both are orthogonal to HilbertA; in fact, our kernel is already built on a FA-style tiled computation.
>
> To provide a more relevant 2D-aware sparse baseline, we evaluate NATTEN [1], which restricts each token to a k $\times$ k neighborhood. Due to space constraints, we provide [the detailed results here](https://anonymous.4open.science/r/hilberta/6_vlm_tab.md). In summary, this proves HilbertA is a superior choice for 2D data.
>
> ## W5 Visual artifacts
>
> We agree that qualitative inspection is valuable. As in Sec. 4.2, tile-boundary artifacts can appear, but they are localized and occur rarely. To provide a comprehensive visual assessment, we have added two sets of uncherry-picked results:
>
> - [50 additional COCO-prompted samples](https://anonymous.4open.science/r/hilberta/1_coco50_vs_flux_grid.md);
> - [200 real-world user-prompted samples](https://anonymous.4open.science/r/hilberta/2_t2iprompt200.md) drawn from t2i-prompts-3m.
>
> Also, no systematic artifacts on other models. [Samples from *Flux.2-Klein*](https://anonymous.4open.science/r/hilberta/3_flux2_comparison.md),
>
> These confirm that HilbertA preserves visual fidelity across different model architectures.
>
> ---
>
> Refernces
>
> [1] Hassani, Ali, et al. "Neighborhood attention transformer." Proceedings of the ICCV. 2023.

---

### Decision · Program_Chairs · 2026-04-30

**Decision:**

Accept (regular)

**Comment:**

The paper received all positive recommendations from four reviewers, with three weak accepts and one accept. The reviewers expressed their support for the acceptance of the paper, especially pointing out the strengths of the paper from different aspects, including the interesting motivation, the novel designs in the overall framework, and the comprehensive experimental verifications. AC agrees with the ratings from the reviewers and recommends acceptance for this submission.